# Bayesian Parameter Shift Rule in Variational Quantum Eigensolvers

**Samuele Pedrielli**[1,2,3*], **Christopher J. Anders**[4], **Lena Funcke**[5,6]
**Karl Jansen**[7,8], **Kim A. Nicoli,**[9,5,6], **Shinichi Nakajima**[1,2,4*]

[1]BIFOLD, Germany, [2] Technische Universität Berlin, Germany
[3]Università degli Studi di Padova, Italy, [4]RIKEN Center for AIP, Japan
[5]Transdisciplinary Research Area (TRA) Matter, University of Bonn, Germany
[6]Helmholtz Institute for Radiation and Nuclear Physics (HISKP), University of Bonn, Germany
[7]Deutsches Elektronen-Synchrotron (DESY), Germany
[8]Computation-Based Science and Technology Research Center, The Cyprus Institute, Cyprus
[9]Oldendorff Carriers GmbH & Co. KG, Germany

## Abstract

*Parameter shift rules* (PSRs) are key techniques for efficient gradient estimation in variational quantum eigensolvers (VQEs). In this paper, we propose their Bayesian variant, where Gaussian processes with appropriate kernels are used to estimate the gradient of the VQE objective. Our *Bayesian PSR* offers flexible gradient estimation from observations at arbitrary locations with uncertainty information, and reduces to the generalized PSR in special cases. In stochastic gradient descent (SGD), the flexibility of Bayesian PSR allows reuse of observations in previous steps, which accelerates the optimization process. Furthermore, the accessibility to the posterior uncertainty, along with our proposed notion of *gradient confident region* (GradCoRe), enables us to minimize the observation costs in each SGD step. Our numerical experiments show that the VQE optimization with Bayesian PSR and GradCoRe significantly accelerates SGD, and outperforms the state-of-the-art methods, including sequential minimal optimization.

## 1 Introduction

The variational quantum eigensolver (VQE) (Peruzzo et al., 2014; McClean et al., 2016) is a hybrid quantum-classical algorithm for approximating the ground state of the Hamiltonian of a given physical system. The quantum part of VQEs uses parameterized quantum circuits to generate trial quantum states and measures the expectation value of the Hamiltonian, i.e., the energy, while the classical part performs energy minimization with noisy observations from the quantum device. Provided that the parameterized quantum circuits can accurately approximate the ground state, the minimized energy gives a tight upper bound of the ground state energy of the Hamiltonian.

The observation noise in the quantum device comes from multiple sources. One source of noise is *measurement shot noise*, which arises from the statistical nature of quantum measurements—outcomes follow the probabilities specified by the quantum state, and finite sampling introduces fluctuations. Since this noise source is random and independent, it can be reduced by increasing the number of measurement shots, to which the variance is inversely proportional. Another source of noise stems from imperfections in the quantum hardware, which have been reduced in recent years by hardware design (Bluvstein et al., 2023), as well as error mitigation (Cai et al., 2023), quantum error correction (Roffe, 2019; Acharya et al., 2024), and machine learning (Liao et al., 2024; Nicoli et al., 2025) techniques. In this paper, we do not consider hardware noise, as is common in papers developing optimization methods (Nakanishi et al., 2020; Nicoli et al., 2023b).

Stochastic gradient descent (SGD), sequential minimal optimization (SMO), and Bayesian optimization (BO) have previously been used to minimize the VQE objective function. Under some mild

---

*Correspondence to `{samuele.pedrielli@campus.,nakajima@}tu-berlin.de`

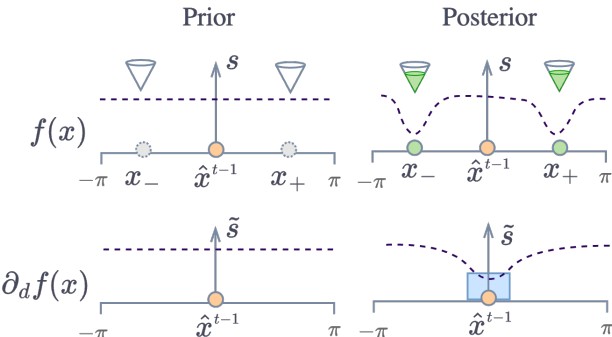

Figure 1: Illustration of our gradient confident region (GradCoRe) approach. Our goal is to minimize the true energy $f^*(\boldsymbol{x})$ over the set of parameters $\boldsymbol{x} \in [0, 2\pi)^D$, where we use a GP surrogate $f(\boldsymbol{x})$ for approximating $f^*(\boldsymbol{x})$. Observing $f^*$ at points $\boldsymbol{x}_-$ and $\boldsymbol{x}_+$ (green circles) along the $d$-th direction (solid horizontal line) decreases the uncertainty (dashed curves) not only for predicting $f(\boldsymbol{x}_{\pm})$, but also for predicting $\partial_d f(\hat{\boldsymbol{x}}^{t-1})$, so that the current optimal point $\hat{\boldsymbol{x}}^{t-1}$ falls within the GradCoRe (blue square). Our GradCoRe-based SGD uses the minimum number of measurement shots for achieving required gradient estimation accuracy in each iteration, and thus minimizes the total observation costs over the optimization process.

assumptions (Nakanishi et al., 2020), this objective function is known to have special properties. Based on those properties, SGD methods can use the gradient estimated by so-called *parameter shift rules* (PSRs) (Mitarai et al., 2018), and specifically designed SMO (Platt, 1998) methods, called Nakanishi-Fuji-Todo (NFT) (Nakanishi et al., 2020), perform one-dimensional subspace optimization with only a few observations in each iteration. Iannelli and Jansen (2021) applied BO to solve VQEs as noisy global optimization problems.

Although Gaussian processes (GPs) (Rasmussen and Williams, 2006) have been used in VQEs as common surrogate functions for BO (Frazier, 2018), they have also been used to improve SGD-based and SMO-based methods. Nicoli et al. (2023a) proposed the *VQE kernel*—a physics-informed kernel that fully reflects the properties of VQEs—and combined SMO and BO with the *expected maximum improvement within confident region* (EMICoRe) acquisition function. This allows for identification of the optimal locations to measure on the quantum computer in each SMO iteration. Tamiya and Yamasaki (2022) combined SGD and BO, and proposed *stochastic gradient line BO* (SGLBO), which uses BO to identify the optimal step size in each SGD iteration. Anders et al. (2024) proposed the *subspace in confident region* (SubsCoRe) approach, where the observation costs are minimized based on the posterior uncertainty estimation in each SMO iteration.

In this paper, we take a different approach to leveraging GPs, and introduce a *Bayesian parameter shift rule* (Bayesian PSR), where the gradient of the VQE objective is estimated using GPs with the VQE kernel. The Bayesian PSR translates into a regularized variant of PSRs if the observations are performed at designated locations. However, our approach offers significant advantages—flexibility and direct access to uncertainty—over existing PSRs (Mitarai et al., 2018; Wierichs et al., 2022). More specifically, the Bayesian PSR can use observations at any set of locations, which allows the reuse of observations performed in previous iterations of SGD. Reusing previous observations along with new observations improves the gradient estimation accuracy, and thus accelerates the optimization process. Furthermore, the uncertainty information can be used to adapt the observation cost in each SGD iteration—in a similar spirit to Anders et al. (2024)—which significantly reduces the cost of obtaining new observations, while maintaining a required level of accuracy. We implement this adaptive observation cost strategy by introducing a novel notion of *gradient confidence region* (GradCoRe)—the region in which the uncertainty of the gradient estimation is below a specified threshold (see Figure 1). Empirical evaluations show that our proposed Bayesian PSR improves the gradient estimator, and SGD equipped with our GradCoRe approach outperforms all previous state-of-the-art methods including NFT and its variants.

The main contributions are summarized as follows:

- We propose *Bayesian PSR*, a flexible variant of existing PSRs that provides access to uncertainty information.

- We theoretically establish the relationship between Bayesian PSR and existing PSRs, revealing the optimality of the *shift* parameter in first-order PSRs.

- We introduce the notion of *GradCoRe*, and propose an adaptive observation cost strategy for SGD optimization.

- We numerically validate our theory and empirically demonstrate the effectiveness of the proposed Bayesian PSR and GradCoRe.

**Related work:** Finding the optimal set of parameters for a variational quantum circuit is a challenging problem, prompting the development of various approaches to improve the optimization in VQEs. Gradient-based methods for VQEs often rely on PSRs (Mitarai et al., 2018; Wierichs et al., 2022), which enable reasonably accurate gradient estimation of the output of quantum circuits with respect to their parameters. Nakanishi et al. (2020) proposed an SMO (Platt, 1998) algorithm, known as *NFT*, where, at each step of SMO, one parameter is analytically minimized by performing a few observations. Nicoli et al. (2023a) combined NFT with GP and BO by developing a physics-inspired kernel for GP regression and proposing the EMICoRe acquisition function, relying on the concept of confident regions (CoRe). This method improves upon NFT by leveraging the information from observations in previous steps to identify the optimal locations to perform the next observations. Anders et al. (2024) leveraged the same notion of CoRe, and proposed SubsCoRe, where, instead of optimizing the observed locations, the minimal number of measurement shots is identified to achieve the required accuracy defined by the CoRe. The resulting algorithm converges to the same energy as NFT with a smaller quantum computation cost, i.e., the total number of measurement shots on a quantum computer. Tamiya and Yamasaki (2022) combined SGD with BO to tackle the excessive cost of standard SGD approaches and used BO to accelerate the convergence by finding the optimal step size. In a general context of BO, Müller et al. (2021) proposed a gradient information with BO (GIBO) approach, where the uncertainty of the GP-estimated gradient is minimized. Our GradCoRe can be seen as an enhanced version of GIBO, where the theoretically optimal locations are observed with minimum costs based on strong physical information of VQEs.

## 2 BACKGROUND

Here we briefly introduce Gaussian process (GP) regression and its derivatives, as well as VQEs with their known properties.

### 2.1 GP REGRESSION AND DERIVATIVE GP

Assume that we aim to learn an unknown function $f^*(\cdot) : \mathcal{X} \mapsto \mathbb{R}$ from the training data $\boldsymbol{X} = (\boldsymbol{x}_1, \ldots, \boldsymbol{x}_N) \in \mathcal{X}^N, \boldsymbol{y} = (y_1, \ldots, y_N)^\top \in \mathbb{R}^N, \boldsymbol{\sigma} = (\sigma_1^2, \ldots, \sigma_N^2) \in \mathbb{R}_{++}^N$ that fulfills

$$y_n = f^*(\boldsymbol{x}_n) + \varepsilon_n, \qquad\qquad \varepsilon_n \sim \mathcal{N}_1(0, \sigma_n^2), \qquad\qquad (1)$$

where $\mathcal{N}_D(\cdot; \boldsymbol{\mu}, \boldsymbol{\Sigma})$ denotes the $D$-dimensional Gaussian distribution with mean $\boldsymbol{\mu}$ and covariance $\boldsymbol{\Sigma}$. With the Gaussian process (GP) prior $p(f(\cdot)) = \mathrm{GP}(f(\cdot); 0(\cdot), k(\cdot, \cdot))$, where $0(\cdot)$ and $k(\cdot, \cdot)$ are the prior zero-mean and the kernel (covariance) functions, respectively, the posterior distribution of the function values $\boldsymbol{f}' = (f(\boldsymbol{x}'_1), \ldots, f(\boldsymbol{x}'_M))^\top \in \mathbb{R}^M$ at arbitrary test points $\boldsymbol{X}' = (\boldsymbol{x}'_1, \ldots, \boldsymbol{x}'_M) \in \mathcal{X}^M$ is given as

$$p(\boldsymbol{f}'|\boldsymbol{X}, \boldsymbol{y}) = \mathcal{N}_M(\boldsymbol{f}'; \boldsymbol{\mu}'_{[\boldsymbol{X}, \boldsymbol{y}, \boldsymbol{\sigma}]}, \boldsymbol{S}'_{[\boldsymbol{X}, \boldsymbol{\sigma}]}), \quad \text{where} \qquad\qquad (2)$$

$$\boldsymbol{\mu}'_{[\boldsymbol{X}, \boldsymbol{y}, \boldsymbol{\sigma}]} = \boldsymbol{K}'^\top (\boldsymbol{K} + \mathbf{Diag}(\boldsymbol{\sigma}))^{-1} \boldsymbol{y} \quad \text{and} \quad \boldsymbol{S}'_{[\boldsymbol{X}, \boldsymbol{\sigma}]} = \boldsymbol{K}'' - \boldsymbol{K}'^\top (\boldsymbol{K} + \mathbf{Diag}(\boldsymbol{\sigma}))^{-1} \boldsymbol{K}' \quad (3)$$

are the posterior mean and covariance, respectively (Rasmussen and Williams, 2006). Here $\mathbf{Diag}(\boldsymbol{v})$ is the diagonal matrix with $\boldsymbol{v}$ specifying the diagonal entries, and $\boldsymbol{K} = k(\boldsymbol{X}, \boldsymbol{X}) \in \mathbb{R}^{N \times N}, \boldsymbol{K}' = k(\boldsymbol{X}, \boldsymbol{X}') \in \mathbb{R}^{N \times M}$, and $\boldsymbol{K}'' = k(\boldsymbol{X}', \boldsymbol{X}') \in \mathbb{R}^{M \times M}$ are the train, train-test, and test kernel matrices, respectively, where $k(\boldsymbol{X}, \boldsymbol{X}')$ denotes the kernel matrix evaluated at each column of $\boldsymbol{X}$ and $\boldsymbol{X}'$ such that $(k(\boldsymbol{X}, \boldsymbol{X}'))_{n,m} = k(\boldsymbol{x}_n, \boldsymbol{x}'_m)$. We also denote the posterior as $p(f(\cdot)|\boldsymbol{X}, \boldsymbol{y}) = \mathrm{GP}(f(\cdot); \mu_{[\boldsymbol{X}, \boldsymbol{y}, \boldsymbol{\sigma}]}(\cdot), s_{[\boldsymbol{X}, \boldsymbol{\sigma}]}(\cdot, \cdot))$ with the posterior mean $\mu_{[\boldsymbol{X}, \boldsymbol{y}, \boldsymbol{\sigma}]}(\cdot)$ and covariance $s_{[\boldsymbol{X}, \boldsymbol{\sigma}]}(\cdot, \cdot)$ functions.

Since the derivative operator is linear, the derivative $\boldsymbol{\nabla_x} f = (\partial_1 f, \ldots, \partial_D f)^\top \in \mathbb{R}^D$ of GP samples also follows a GP. Here we abbreviate $\partial_d = \frac{\partial}{\partial x_d}$. Since the kernel function corresponds to the

covariance of GP prior, we can straightforwardly handle the derivative outputs by adjusting the kernel so that it is consistent with the original kernel defined for non-derivative outputs. Assume that $\boldsymbol{x}$ is a training or test point with non-derivative output $y = f^*(\boldsymbol{x}) + \varepsilon$, and $\boldsymbol{x}'$ and $\boldsymbol{x}''$ are training or test points with derivative outputs, $y' = \partial_{d'} f^*(\boldsymbol{x}') + \varepsilon'$, $y'' = \partial_{d''} f^*(\boldsymbol{x}'') + \varepsilon''$. Then, the kernel entries involving those three points should be replaced with

$$\widetilde{k}(\boldsymbol{x}, \boldsymbol{x}') = \text{cov}(f(\boldsymbol{x}), \partial_{d'} f(\boldsymbol{x}')) = \tfrac{\partial}{\partial x'_{d'}} \text{cov}(f(\boldsymbol{x}), f(\boldsymbol{x}')) = \tfrac{\partial}{\partial x'_{d'}} k(\boldsymbol{x}, \boldsymbol{x}'), \tag{4}$$

$$\widetilde{k}(\boldsymbol{x}', \boldsymbol{x}'') = \text{cov}(\partial_{d'} f(\boldsymbol{x}'), \partial_{d''} f(\boldsymbol{x}'')) = \tfrac{\partial^2}{\partial x'_{d'} \partial x''_{d''}} \text{cov}(f(\boldsymbol{x}'), f(\boldsymbol{x}'')) = \tfrac{\partial^2}{\partial x'_{d'} \partial x''_{d''}} k(\boldsymbol{x}', \boldsymbol{x}''). \tag{5}$$

The posterior (2) with appropriately replaced kernel entries gives the posterior distribution of derivatives at test points. We denote the GP posterior of a single component of the derivative as

$$p(\partial_d f(\cdot) | \boldsymbol{X}, \boldsymbol{y}) = \text{GP}\left(\partial_d f(\cdot); \widetilde{\mu}^{(d)}_{[\boldsymbol{X}, \boldsymbol{y}, \boldsymbol{\sigma}]}(\cdot), \widetilde{s}^{(d)}_{[\boldsymbol{X}, \boldsymbol{\sigma}]}(\cdot, \cdot)\right) \tag{6}$$

with the posterior mean $\widetilde{\mu}^{(d)}(\cdot)$ and covariance $\widetilde{s}^{(d)}(\cdot, \cdot)$ functions for the derivative with respect to $x_d$. More generally, GP regression can be analytically performed in the case where the training outputs (i.e., observations) and the test outputs (i.e., predictions) contain derivatives with different orders (see Appendix A for more details).

## 2.2 VARIATIONAL QUANTUM EIGENSOLVERS AND THEIR PHYSICAL PROPERTIES

The VQE (Peruzzo et al., 2014; McClean et al., 2016) is a hybrid quantum-classical computing protocol for estimating the ground-state energy of a given quantum Hamiltonian for a $Q$-qubit system. The quantum computer is used to prepare a parametric quantum state $|\psi_{\boldsymbol{x}}\rangle$, which depends on $D$ angular parameters $\boldsymbol{x} \in \mathcal{X} = [0, 2\pi)^D$. This trial state $|\psi_{\boldsymbol{x}}\rangle$ is generated by applying $D'(\geq D)$ *quantum gate operations*, $G(\boldsymbol{x}) = G_{D'} \circ \cdots \circ G_1$, to an initial quantum state $|\psi_0\rangle$, i.e., $|\psi_{\boldsymbol{x}}\rangle = G(\boldsymbol{x})|\psi_0\rangle$. All gates $\{G_{d'}\}_{d'=1}^{D'}$ are unitary operators, parameterized by at most one variable $x_d$. Let $d(d') : \{1, \ldots, D'\} \mapsto \{1, \ldots, D\}$ be the mapping specifying which one of the variables $\{x_d\}$ parameterizes the $d'$-th gate. We consider parametric gates of the form $G_{d'}(x) = U_{d'}(x_{d(d')}) = \exp\left(-ix_{d(d')}P_{d'}/2\right)$, where $P_{d'}$ is an arbitrary sequence of the Pauli operators $\{\mathbf{1}_q, \sigma_q^X, \sigma_q^Y, \sigma_q^Z\}_{q=1}^Q$ acting on each qubit at most once. This general structure covers both single-qubit gates, such as $R_X(x) = \exp\left(-i\theta\sigma_q^X\right)$, and entangling gates acting on multiple qubits simultaneously, such as $R_{XX}(x) = \exp\left(-ix\sigma_{q_1}^X \circ \sigma_{q_2}^X\right)$ for $q_1 \neq q_2$, commonly realized in trapped-ion quantum hardware setups (Kielpinski et al., 2002; Debnath et al., 2016).

The quantum computer is used to evaluate the energy of the resulting quantum state $|\psi_{\boldsymbol{x}}\rangle$ by observing

$$y = f^*(\boldsymbol{x}) + \varepsilon, \qquad \text{where} \qquad f^*(\boldsymbol{x}) = \langle\psi_{\boldsymbol{x}}|H|\psi_{\boldsymbol{x}}\rangle = \langle\psi_0|G(\boldsymbol{x})^\dagger HG(\boldsymbol{x})|\psi_0\rangle, \tag{7}$$

and $\dagger$ denotes the Hermitian conjugate. For each observation, repeated measurements, called *shots*, on the quantum computer are performed. Averaging over the number $N_{\text{shots}}$ of shots suppresses the variance $\sigma^{*2}(N_{\text{shots}}) \propto N_{\text{shots}}^{-1}$ of the observation noise $\varepsilon$.[1] Since the observation $y$ is the sum of many random variables, it approximately follows the Gaussian distribution, according to the central limit theorem. The Gaussian likelihood (1) therefore approximates the observation $y$ well if $\sigma_n^2 \approx \sigma^{*2}(N_{\text{shots}})$. Using the noisy estimates of $f^*(\boldsymbol{x})$ obtained from the quantum computer, a protocol running on a classical computer is used to solve the following minimization problem:

$$\min_{\boldsymbol{x} \in [0, 2\pi)^D} f^*(\boldsymbol{x}), \tag{8}$$

thus finding the minimizer $\widehat{\boldsymbol{x}}$, i.e., the optimal parameters for the (rotational) quantum gates. Given the high expense of quantum computing resources, the computation cost is primarily driven by quantum operations. As a result, the optimization cost in VQE is typically measured by the total number of measurement shots required during the optimization process.[2] We refer to Tilly et al. (2022) for further details about VQEs and their challenges.

---

[1] We do not consider the hardware noise, and therefore, the observation noise $\varepsilon$ consists only of the *measurement shot* noise.

[2] When the Hamiltonian consists of $N_{\text{og}}$ groups of non-commuting operators, each of which needs to be measured separately, $N_{\text{shots}}$ denotes the number of shots *per operator group*. Therefore, the number of shots *per observation* is $N_{\text{og}} \times N_{\text{shots}}$. In our experiments, we report on the total number of shots per operator group, i.e., the cumulative sum of $N_{\text{shots}}$ over all observations, when evaluating the observation cost.

Let $V_d$ be the number of gates parameterized by $x_d$, i.e., $V_d = |\{d' \in \{1, \ldots D'\}; d = d(d')\}|$. Mitarai et al. (2018) proved that the VQE objective (7) for $V_d = 1$ satisfies the parameter shift rule (PSR)

$$\partial_d f^*(\boldsymbol{x}') = \frac{f^*(\boldsymbol{x}' + \alpha \boldsymbol{e}_d) - f^*(\boldsymbol{x}' - \alpha \boldsymbol{e}_d)}{2 \sin \alpha}, \qquad \forall \boldsymbol{x} \in [0, 2\pi)^D, \ d = 1, \ldots, D, \ \alpha \in [0, 2\pi), \quad (9)$$

where $\{\boldsymbol{e}_d\}_{d=1}^D$ are the standard basis, and the *shift* $\alpha$ is typically set to $\frac{\pi}{2}$. Wierichs et al. (2022) generalized the PSR (9) for arbitrary $V_d$ with equidistant observations $\{\boldsymbol{x}_w = \boldsymbol{x}' + \frac{2w+1}{2V_d} \pi \boldsymbol{e}_d\}_{w=0}^{2V_d - 1}$:

$$\partial_d f^*(\boldsymbol{x}') = \frac{1}{2V_d} \sum_{w=0}^{2V_d - 1} \frac{(-1)^w f^*(\boldsymbol{x}_w)}{2 \sin^2 \left( \frac{(2w+1)\pi}{4V_d} \right)}. \tag{10}$$

Most gradient-based approaches rely on those PSRs, which allow reasonably accurate gradient estimation from $\sum_{d=1}^D 2V_d$ observations. Let

$$\boldsymbol{\psi}_\gamma(\theta) = (\gamma, \sqrt{2} \cos \theta, \sqrt{2} \cos 2\theta, \ldots, \sqrt{2} \cos V_d \theta, \sqrt{2} \sin \theta, \sqrt{2} \sin 2\theta, \ldots, \sqrt{2} \sin V_d \theta)^\top \quad (11)$$

be the (1-dimensional) $V_d$-th order Fourier basis for arbitrary $\gamma > 0$. Nakanishi et al. (2020) found that the VQE objective function $f^*(\cdot)$ in Eq. (7) with any[3] $G(\cdot)$, $H$, and $|\psi_0\rangle$ can be expressed exactly as

$$f^*(\boldsymbol{x}) = \boldsymbol{b}^\top \mathbf{vec} \left( \otimes_{d=1}^D \boldsymbol{\psi}_\gamma(x_d) \right) \tag{12}$$

for some $\boldsymbol{b} \in \mathbb{R}^{\prod_{d=1}^D (1 + 2V_d)}$, where $\otimes$ and $\mathbf{vec}(\cdot)$ denote the tensor product and the vectorization operator for a tensor, respectively. Based on this property, the Nakanishi-Fuji-Todo (NFT) method (Nakanishi et al., 2020) performs SMO (Platt, 1998), where the optimum in a chosen 1D subspace for each iteration is analytically estimated from only $1 + 2V_d$ observations (see Appendix B for the detailed procedure). It was shown that the PSR (9) and the trigonometric polynomial function form (12) are mathematically equivalent (Nicoli et al., 2023a).

Inspired by the function form (12) of the objective, Nicoli et al. (2023a) proposed the VQE kernel

$$k_\gamma(\boldsymbol{x}, \boldsymbol{x}') = \sigma_0^2 \prod_{d=1}^D \left( \frac{\gamma^2 + 2 \sum_{v=1}^{V_d} \cos\left(v(x_d - x_d')\right)}{\gamma^2 + 2V_d} \right), \tag{13}$$

which is decomposed as $k_\gamma(\boldsymbol{x}, \boldsymbol{x}') = \boldsymbol{\phi}_\gamma(\boldsymbol{x})^\top \boldsymbol{\phi}_\gamma(\boldsymbol{x}')$ with feature maps $\boldsymbol{\phi}_\gamma(\boldsymbol{x}) = \frac{\sigma_0}{(\gamma^2 + 2V_d)^{D/2}} \mathbf{vec} \left( \otimes_{d=1}^D \boldsymbol{\psi}_\gamma(x_d) \right)$, for GP regression. The kernel parameter $\gamma^2$ controls the smoothness of the function, i.e., suppressing the interaction terms when $\gamma^2 > 1$. When $\gamma^2 = 1$, the Fourier basis (11) is orthonormal, and the VQE kernel (13) is proportional to the product of Dirichlet kernels (Rudin, 1964). The VQE kernel reflects the physical knowledge (12) of VQE, and thus allows us to perform a Bayesian variant of NFT—*Bayesian NFT* or *Bayesian SMO*—where the 1D subspace optimzation in each SMO step is performed with GP (see Appendix B for more details and the performance comparison between the original NFT and Bayesian NFT). Nicoli et al. (2023a) furthermore enhanced Bayesian NFT with BO, using the notion of confident region (CoRe),

$$\mathcal{Z}_{[\boldsymbol{X}, \boldsymbol{\sigma}]}(\kappa^2) = \left\{ \boldsymbol{x} \in \mathcal{X}; s_{[\boldsymbol{X}, \boldsymbol{\sigma}]}(\boldsymbol{x}, \boldsymbol{x}) \leq \kappa^2 \right\}, \tag{14}$$

i.e., the region in which the uncertainty of the GP prediction is lower than a threshold $\kappa$. More specifically, they introduced the EMICoRe acquisition function to find the best observation points in each SMO iteration, such that the maximum expected improvement within the CoRe is maximized.

## 3  BAYESIAN PARAMETER SHIFT RULES

We propose *Bayesian PSR*, which estimates the gradient of the VQE objective (7) by the GP posterior (6) with the VQE kernel (13) along with its derivatives (4) and (5) (which can be explicitly given as Eqs.(38) and (39) in Appendix C.3). The advantages of Bayesian PSR include: 1) The gradient estimator has an analytic-form, 2) Estimation can be performed using observations at any set of points, 3) Estimation is optimal for heteroschedastically noisy observations (from the Bayesian perspective), as long as the prior with the kernel parameters, $\gamma$ and $\sigma_0^2$, is appropriately set, and 4) The posterior

---

[3]Any circuit consisting of parametrized rotation gates and non-parametric unitary gates.

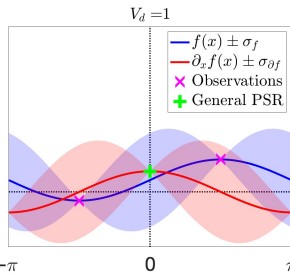 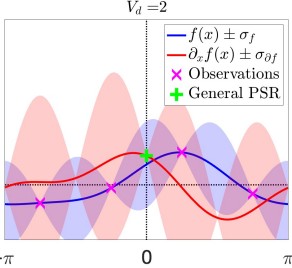 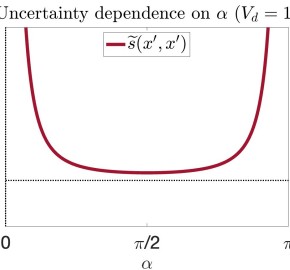

Figure 2: Illustration of the behavior of the Bayesian PSR when $V_d = 1$ (left) and when $V_d = 2$ (middle). Bayesian PSR prediction (red) coincides with general PSR (green cross) for the designed equidistant observations (magenta crosses). The right plot visualizes the variance (the second equation in Eq. (16)) of the derivative GP prediction at $x'$, as a function of the shift $\alpha$ of observations when $V_d = 1$. Intuitively, the minimum uncertainty is achieved with $\alpha = \frac{\pi}{2}$, which corresponds to the maximum span ($= \pi$) between the two observed points. For all panels, the noise and kernel parameters are set to $\sigma^2 = 0.01, \gamma^2 = 9, \sigma_0^2 = 100$.

uncertainty can be analytically computed *before* performing the observations. In Section 4, we propose novel SGD solvers for VQEs that leverage the advantages of Bayesian PSR.

As naturally expected, our Bayesian PSR is a generalization of exisiting PSRs, and reduces to the general PSR (10) for noiseless and equidistant observations. Let $\mathbf{1}_D \in \mathbb{R}^D$ be the vector with all entries equal to one.

**Theorem 3.1.** *For any $x' \in [0, 2\pi)^D$ and $d = 1, \ldots, D$, the mean and variance of the derivative GP prediction, given observations $\boldsymbol{y} = (y_0, \ldots, y_{2V_d-1})^\top \in \mathbb{R}^{2V_d}$ at $2V_d$ equidistant training points $\boldsymbol{X} = (\boldsymbol{x}_0, \ldots, \boldsymbol{x}_{2V_d-1}) \in \mathbb{R}^{D \times 2V_d}$ for $\boldsymbol{x}_w = \boldsymbol{x}' + \frac{2w+1}{2V_d}\pi \boldsymbol{e}_d$ with homoschedastic noise $\boldsymbol{\sigma} = \sigma^2 \cdot \mathbf{1}_{2V_d}$ for $\sigma^2 \ll \sigma_0$, are*

$$\widetilde{\mu}_{[\boldsymbol{X},\boldsymbol{y},\boldsymbol{\sigma}]}^{(d)}(\boldsymbol{x}') = \frac{\sum_{w=0}^{2V_d-1} \frac{(-1)^w y_w}{2\sin^2\left(\frac{(2w+1)\pi}{4V_d}\right)}}{(\gamma^2 + 2V_d)\frac{\sigma^2}{\sigma_0^2} + 2V_d} + O(\frac{\sigma^4}{\sigma_0^4}), \quad \widetilde{s}_{[\boldsymbol{X},\boldsymbol{\sigma}]}^{(d)}(\boldsymbol{x}', \boldsymbol{x}') = \sigma^2 \left(\frac{2V_d^2+1}{6}\right) + O(\frac{\sigma^4}{\sigma_0^2}). \quad (15)$$

The proof, the non-asymptotic form of the mean and the variance, and the numerical validation of the theorem are given in Appendix C. Apparently, the mean prediction (the first equation in Eq. (15)) by Bayesian PSR converges to the general PSR (10) with the uncertainty (the second equation in Eq. (15)) converging to zero in the noiseless limit, i.e., $\sigma^2 \to +0$ and hence $y_w = f^*(\boldsymbol{x}_w)$. In noisy cases, the prior variance $\sigma_0^2 \sim O(\sigma^2)$ suppresses the amplitude of the gradient estimator as a regularizer through the first term in the denominator in the first equation of Eq. (15).

Figure 2 illustrates the behavior of Bayesian PSR when $V_d = 1$ (left panel) and when $V_d = 2$ (middle panel). In each panel, given $2V_d$ equidistant observations (magenta crosses), the blue curve shows the (non-derivative) GP prediction with uncertainty (blue shades), while the red curve shows the derivative GP prediction with uncertainty (red shades). Note the $\frac{\pi}{2V_d}$ shift of the low uncertainty locations between the GP prediction (blue) and the derivative GP prediction (red). The green cross shows the output of the general PSR (10) at $\boldsymbol{x}' = 0$, which almost coincides with the Bayesian PSR prediction (red curve) under this setting. Other examples, including cases where the Bayesian regularization is visible, are given in Appendix C.

In the simplest first-order case, i.e., where $V_d = 1, \forall d = 1, \ldots, D$, we can theoretically investigate the optimality of the choice of the shift $\alpha$ in Eq. (9) (the proof is also given in Appendix C).

**Theorem 3.2.** *Assume that $V_d = 1, \forall d = 1, \ldots, D$. For any $x' \in [0, 2\pi)^D$ and $d = 1, \ldots, D$, the mean and variance of the derivative GP prediction, given observations $\boldsymbol{y} = (y_1, y_2)^\top \in \mathbb{R}^2$ at two training points $\boldsymbol{X} = (\boldsymbol{x}' - \alpha \boldsymbol{e}_d, \boldsymbol{x}' + \alpha \boldsymbol{e}_d) \in \mathbb{R}^{D \times 2}$ with homoschedastic noise $\boldsymbol{\sigma} = (\sigma^2, \sigma^2)^\top$, are*

$$\widetilde{\mu}_{[\boldsymbol{X},\boldsymbol{y},\boldsymbol{\sigma}]}^{(d)}(\boldsymbol{x}') = \frac{(y_2 - y_1)\sin\alpha}{(\gamma^2/2+1)\sigma^2/\sigma_0^2 + 2\sin^2\alpha}, \quad \widetilde{s}_{[\boldsymbol{X},\boldsymbol{\sigma}]}^{(d)}(\boldsymbol{x}', \boldsymbol{x}') = \frac{\sigma^2}{(\gamma^2/2+1)\sigma^2/\sigma_0^2 + 2\sin^2\alpha}. \quad (16)$$

Again, the mean prediction (the first equation in Eq. (16)) is a regularized version of the PSR (9). The uncertainty prediction (the second equation in Eq. (16)) implies that $\alpha = \pi/2$ minimizes the

uncertainty in the noisy case, regardless of $\sigma^2, \sigma_0^2$ and $\gamma$ (see the right panel in Figure 2, where the variance of the derivative GP prediction at $\boldsymbol{x}'$ is visualized as a function of the shift $\alpha$ of observations for $V_d = 1$). This supports most of the use cases of the PSR in the literature (Mitarai et al., 2018), and matches the intuition that the maximum span minimizes the uncertainty.

## 4    SGD WITH BAYESIAN PSR

In this section, we equip SGD with Bayesian PSR. In the standard implementation of SGD for VQEs, $2V_d$ equidistant points along each direction $d = 1, \ldots, D$ are observed for gradient estimation by the general PSR (10) (or by the PSR (9) if $V_d = 1, \forall d$) in each SGD iteration.

**Bayesian SGD (Bayes-SGD):**    A straightforward application of Bayesian PSR is to replace existing PSRs with Bayesian PSR for gradient estimation, allowing for the reuse of previous observations. We retain $R \cdot 2V_d \cdot D$ latest observations for a predetermined $R$ in our experiments. Reusing previous observations accumulates the gradient information, and thus improves the gradient estimation accuracy, as shown in Section 5.2.

### 4.1    GRADIENT CONFIDENT REGION (GRADCORE)

We propose an adaptive observation cost control strategy that leverages the uncertainty information provided by the Bayesian PSR. This strategy adjusts the number of measurement shots for gradient estimation in each SGD iteration so that the variances of the derivative GP prediction at the current optimal point $\widehat{\boldsymbol{x}}$ are below certain thresholds. In a similar fashion to the CoRe (14), we define the *gradient confident region* (GradCoRe)

$$\widetilde{\mathcal{Z}}_{[\boldsymbol{X}, \boldsymbol{\sigma}]}(\boldsymbol{\kappa}) = \left\{ \boldsymbol{x} \in \mathcal{X} ; \widetilde{s}_{[\boldsymbol{X}, \boldsymbol{\sigma}]}^{(d)}(\boldsymbol{x}, \boldsymbol{x}) \leq \kappa_d^2, \forall d \right\}, \tag{17}$$

where $\boldsymbol{\kappa} = (\kappa_1^2, \ldots, \kappa_D^2)^\top \in \mathbb{R}^D$ are the required accuracy thresholds. Our proposed SGD-based optimizer, named *SGD-GradCoRe*, measures new equidistant points $\breve{\boldsymbol{X}} = \{\{\boldsymbol{x}_w^{(d)} = \widehat{\boldsymbol{x}} + \frac{2w+1}{2V_d}\pi\boldsymbol{e}_d\}_{w=0}^{2V_d}\}_{d=1}^D$ for all directions with the minimum total number of shots such that the current optimal point $\widehat{\boldsymbol{x}}$ is in the GradCoRe (see Figure 1).

Before starting optimization, we evaluate the single-shot observation noise variance $\sigma^{*2}(1) = \overline{\sigma}^{*2}$ by collecting measurements at random locations, following Anders et al. (2024). We use this information to estimate the observation noise variance as a function of the number of shots as $\sigma^{*2}(N_{\text{shots}}) = \frac{\overline{\sigma}^{*2}}{N_{\text{shots}}}$. Let $(\boldsymbol{X}^t, \boldsymbol{y}^t, \boldsymbol{\sigma}^t)$ be the training data (all previous observations) at the $t$-th SGD iteration step, and let $\breve{\boldsymbol{\nu}} \in \mathbb{R}^{2V_d D}$ be the vector of the numbers of measurement shots at the new equidistant measurement points $\breve{\boldsymbol{X}}$ for all directions. Before measuring at $\breve{\boldsymbol{X}}$ in the $(t+1)$-th SGD iteration, we solve the following problem:

$$\min_{\widetilde{\boldsymbol{\nu}}} \|\widetilde{\boldsymbol{\nu}}\|_1 \text{ s.t. } \widehat{\boldsymbol{x}} \in \widetilde{\mathcal{Z}}_{[(\boldsymbol{X}^t, \breve{\boldsymbol{X}}), (\boldsymbol{\sigma}^t, \breve{\boldsymbol{\sigma}}(\widetilde{\boldsymbol{n}}))]}(\boldsymbol{\kappa}(t)), \tag{18}$$

where $\breve{\boldsymbol{\sigma}}(\widetilde{\boldsymbol{\nu}}) = \overline{\sigma}^{*2} \cdot (\widetilde{\nu}_1^{-1}, \ldots, \widetilde{\nu}_{2V_d D}^{-1})^\top$, and $\boldsymbol{\kappa}(t)$ is the required accuracy dependent on the iteration step $t$. Informally, we minimize the total measurement budget under the constraint that the posterior gradient variance along each direction $d$ is smaller than the required accuracy threshold. For simplicity, we solve the GradCoRe problem (18) by grid search under the additional constraint that all $2V_d D$ points are measured with an equal number of shots.

We set the required accuracy thresholds to $\boldsymbol{\kappa}(t) = \kappa^2(t)\mathbf{1}_D$, where

$$\kappa^2(t) = \max\left( c_0, \frac{c_1}{D} \sum_{d=1}^D \left( \widetilde{\mu}_{[\boldsymbol{X}^t, \boldsymbol{y}^t, \boldsymbol{\sigma}^t]}^{(d)}(\widehat{\boldsymbol{x}}^t) \right)^2 \right). \tag{19}$$

Namely, $\kappa(t)$ is set proportional to the L2-norm of the estimated gradient at the current optimal point at the $t$-th SGD iteration, as long as it is larger than a lower bound. The lower bound $c_0$ and the slope $c_1$ are hyperparameters to be tuned. This strategy for setting the required accuracy based on the estimated gradient norm was proposed by Tamiya and Yamasaki (2022).

In the experiment plots in Section 5, we will refer to SGD-GradCoRe as *GradCoRe*. Further algorithmic details, including pseudo-code and used hyperparameter values, are given in Appendix D.

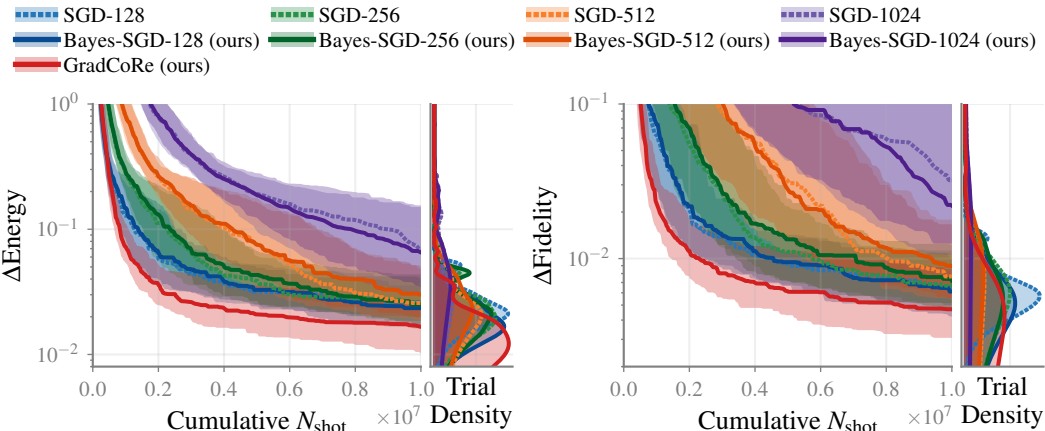

Figure 3: Comparison between SGD with PSR (dashed curves) and SGD with Bayesian PSR (solid curves), as well as GradCoRe (red solid curve), on the Ising Hamiltonian with a $(Q = 5)$-qubits $(L = 3)$-layers quantum circuit. The energy (left) and fidelity (right) are plotted as functions of the cumulative $N_{\text{shots}}$, i.e., the total number of measurement shots. Except GradCoRe equipped with the adaptive shots strategy, the number of shots per observation is set to $N_{\text{shots}} = 128$ (blue), 256 (green), 512 (orange), and 1024 (purple).

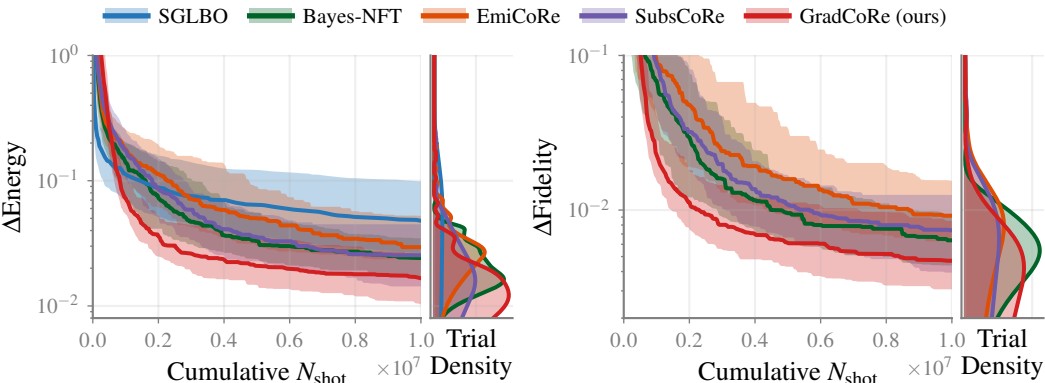

Figure 4: Energy (left) and fidelity (right) achieved within the cumulative number of measurement shots for the Ising Hamiltonian with a $(Q = 5)$-qubits $(L = 3)$-layers quantum circuit. The curves correspond to SGLBO (blue), Bayes-NFT (green), EMICoRe (orange), SubsCoRe (purple), and our proposed GradCoRe (red).

## 5 EXPERIMENTS

### 5.1 SETUP

We demonstrate the performance of our Bayesian PSR and GradCoRe approaches in the same setup used by Nicoli et al. (2023a). For all experiments, we prepared 100 different random initial points, from which all optimization methods start. Our Python implementation uses `Qiskit` (Abraham et al., 2019) for the classical simulation of quantum hardware. The implementation for reproducing our results is attached as supplemental material.

**Hamiltonian and Quantum Circuit:** We focus on the quantum Heisenberg Hamiltonian with open boundary conditions,

$$H = -\sum_{i \in \{X,Y,Z\}} \left[ \sum_{j=1}^{Q-1} (J_i \sigma_j^i \sigma_{j+1}^i) + \sum_{j=1}^{Q} h_i \sigma_j^i \right], \tag{20}$$

where $\{\sigma_j^i\}_{i \in \{X,Y,Z\}}$ are the Pauli operators acting on the $j$-th qubit. For the quantum circuit, we use a common ansatz, called the $L$-layered `Efficient SU(2)` circuit with open boundary conditions, where $V_d = 1, \forall d$ (see Nicoli et al. (2023a) for more details).

**Evaluation Metrics:** We compare all methods using two metrics: the best achieved *true energy* $f^*(\widehat{x})$, for $f^*(\cdot)$ defined in Eq. (7), and *fidelity* $|\langle \psi_{\mathrm{GS}} | \psi_{\widehat{x}} \rangle|^2 \in [0, 1]$. The latter is the inner product between the true ground-state wave function $|\psi_{\mathrm{GS}}\rangle$, computed by exact diagonalization of the target Hamiltonian $H$, and the trial wave function, $|\psi_{\widehat{x}}\rangle$, corresponding to the quantum state generated by the circuit using the optimized parameters $\widehat{x}$. For both metrics, we plot the difference (smaller is better) to the respective target, i.e.,

$$\Delta\mathrm{Energy} = \langle \psi_{\widehat{x}} | H | \psi_{\widehat{x}} \rangle - \langle \psi_{\mathrm{GS}} | H | \psi_{\mathrm{GS}} \rangle = f^*(\widehat{x}) - \langle \psi_{\mathrm{GS}} | H | \psi_{\mathrm{GS}} \rangle, \tag{21}$$

$$\Delta\mathrm{Fidelity} = \langle \psi_{\mathrm{GS}} | \psi_{\mathrm{GS}} \rangle - \langle \psi_{\mathrm{GS}} | \psi_{\widehat{x}} \rangle = 1 - \langle \psi_{\mathrm{GS}} | \psi_{\widehat{x}} \rangle, \tag{22}$$

in log scale. Here, $|\psi_{\mathrm{GS}}\rangle$ and $\langle \psi_{\mathrm{GS}} | H | \psi_{\mathrm{GS}} \rangle$ are the wave function and true energy at the ground-state, respectively, both of which are computed analytically. As a measure of the quantum computation cost, we consider the total number of measurement shots *per operator group* (see Footnote 2) for all observations over the whole optimization process.

**Baseline Methods:** We compare our Bayesian SGD and GradCoRe approaches to the baselines, including SGD with the PSR (9), Bayesian NFT, SGLBO (Tamiya and Yamasaki, 2022), EMICoRe (Nicoli et al., 2023a), and SubsCoRe (Anders et al., 2024). We exclude the original NFT (Nakanishi et al., 2020) because it is outperformed by Bayesian NFT (see Figure 5 in Appendix B). We also exclude GIBO (Müller et al., 2021), which is an even weaker baseline than the original NFT (see Appendix G).

**Algorithm Setting:** All SGD-based methods use the ADAM optimizer with $l_r = 0.05$, $\beta s = (0.9, 0.999)$. For the methods not equipped with adaptive cost control (i.e., all methods except SGLBO, SubsCoRe and GradCoRe), we set $N_{\mathrm{shots}} = 1024$ for each observation—the same setting as in Nicoli et al. (2023a)—unless specified explicitly. To avoid error accumulation, all SMO-based methods measure the "center", i.e., the current optimal point without shift, every $D + 1$ iterations (Nakanishi et al., 2020). Bayes-SGD and GradCoRe estimate the gradient from the $R \cdot 2V_d \cdot D$ latest observations for $R = 5$, and GradCoRe initially uses the fixed threshold $\kappa^2(t) = \overline{\sigma}^{*2}/256$ before starting the cost adaption after $D$ SGD iterations. Further details on the algorithmic and experimental settings are described in Appendix D and Appendix E, respectively.

## 5.2 IMPROVEMENT OVER SGD WITH BAYESIAN PSR AND GRADCORE

First, we investigate how our Bayesian PSR and GradCoRe improve SGD. Figure 3 compares SGD with the standard PSR (SGD) and SGD with Bayesian PSR (Bayes-SGD) on the Ising Hamiltonian, i.e., Eq. (20) for $J_{i \in \{X,Y,Z\}} = (-1, 0, 0)$ and $h_{i \in \{X,Y,Z\}} = (0, 0, -1)$, with a $(Q = 5)$-qubits $(L = 3)$-layers quantum circuit. Both for SGD and Bayes-SGD, the optimization performance with $N_{\mathrm{shots}} = 128, 256, 512, 1024$ measurement shots are shown. The left and right panels plot the difference to the ground-state in true energy (21) and fidelity (22) achieved by each method as functions of the cumulative $N_{\mathrm{shots}}$, i.e., the total number of measurement shots. To the right of each panel, the *trial density*, i.e., the distribution over the trials computed by kernel-density estimation, after the use of $1 \times 10^7$ total measurement shots is depicted. The median, the 25-th and the 75-th percentiles are shown as a solid curve and shades, respectively. We observe that, although Bayesian PSR provides a more accurate gradient estimator, as shown in Figure 7 in Appendix F, the optimization performance is on par with the SGD with the standard PSR. On the other hand, GradCoRe outperforms SGD and Bayes-SGD with different fixed number of shots ($N_{\mathrm{shots}}$) through the entire optimization process. Note that GradCoRe is built on the Bayesian PSR framework, which provides uncertainty estimation as stated in Theorem 3.1. This enables the method to automatically determine the optimal number of measurement shots at each optimization step. The adaptively selected number of shots and the accuracy threshold $\kappa(t)$ for GradCoRe are shown in Appendix F.

## 5.3 COMPARISON WITH STATE-OF-THE-ART METHODS

Figure 4 compares GradCoRe to the baseline methods, SGLBO, Bayes-NFT, EMICoRe, and SubsCoRe. Our GradCoRe, which significantly improves upon SGD as shown in Figure 3, establishes

itself as the new state-of-the-art, exhibiting faster convergence and achieving lower overall energy (see Table 3 in Appendix F.1 for statistical significance test results. We also conducted experiments with different $Q$ and $L$, as well as for the Heisenberg Hamiltonian, on which the results are reported in Appendix F.1.

## 6 CONCLUSION

The physical properties of variational quantum eigensolvers (VQEs) allow us to use specialized optimization methods, i.e., stochastic gradient descent (SGD) with parameter shift rules (PSRs) and a specialized sequential minimal optimization (SMO), called NFT (Nakanishi et al., 2020). Recent research has shown that those properties can be appropriately captured by the physics-informed VQE kernel, with which NFT has been successfully improved through Bayesian machine learning techniques. For instance, observations in previous SMO iterations are used to determine the optimal measurement points (Nicoli et al., 2023a), and observation costs are minimized based on the uncertainty prediction (Anders et al., 2024). In this paper, we have shown that a similar approach can also improve SGD-based methods. Specifically, we proposed Bayesian PSR, where the gradient is estimated by derivative Gaussian processes (GPs). Bayesian PSR generalizes existing PSRs to allow for flexible estimation from observations at an arbitrary set of locations. Furthermore, it provides uncertainty information, which enables observation cost adaptation through the novel notion of gradient confident region (GradCoRe). Our theoretical analysis revealed the relation between Bayesian PSR and existing PSRs, while our numerical investigation empirically demonstrated the utility of our approaches. We envisage that Bayesian approaches will facilitate further development of more efficient algorithms for VQEs and, more generally, quantum computing. In future work, we aim to explore the optimal combination of existing methods and strategies for selecting the most suitable approaches for specific tasks, i.e., specific Hamiltonians.

## ACKNOWLEDGEMENTS

The authors thank the reviewers for their constructive comments and discussion for improving the paper. The authors also thank Stefan Kühn for valuable inputs and discussions. This work was supported by the German Ministry for Education and Research (BMBF) under the grant BIFOLD25B, the European Union's HORIZON MSCA Doctoral Networks programme project AQTIVATE (101072344), the Deutsche Forschungsgemeinschaft (DFG, German Research Foundation) as part of the CRC 1639 NuMeriQS – project no. 511713970 and under Germany's Excellence Strategy – Cluster of Excellence Matter and Light for Quantum Computing (ML4Q) EXC 2004/2 – 390534769, the European Union's Horizon Europe Framework Programme (HORIZON) under the ERA Chair scheme with grant agreement no. 101087126, and the Ministry of Science, Research and Culture of the State of Brandenburg within the Centre for Quantum Technologies and Applications (CQTA). C.J.A. is supported by the Bayes duality project, JST CREST Grant Number JPMJCR2112.

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

## A  GENERAL GAUSSIAN PROCESSES (GPs) WITH DERIVATIVE OUTPUTS

The derivative GP regression can be straightforwardly extended to the case where both training outputs (i.e., observations), and test outputs (i.e., predictions) contain different orders of derivatives.

Assume that we have a set of input points, and for each input point $\boldsymbol{x} \in \mathbb{R}^D$, the corresponding output, i.e., observation or prediction, is $f(\boldsymbol{x})$ or $\partial_{x_d} f(\boldsymbol{x})$, where $\partial_{x_d} \equiv \frac{\partial}{\partial x_d}$. Let us denote the derivative kernel functions as

$$\widetilde{k}^{(d,d')}(\boldsymbol{x}, \boldsymbol{x}') = \begin{cases} k(\boldsymbol{x}, \boldsymbol{x}') & \text{if } d = 0, d' = 0, \\ \partial_{x'_{d'}} k(\boldsymbol{x}, \boldsymbol{x}') & \text{if } d = 0, d' = 1, \ldots, D, \\ \partial_{x_d} k(\boldsymbol{x}, \boldsymbol{x}') & \text{if } d = 1, \ldots, D, d' = 0, \\ \partial_{x_d} \partial_{x'_{d'}} k(\boldsymbol{x}, \boldsymbol{x}') & \text{if } d = 1, \ldots, D, d' = 1, \ldots, D. \end{cases}$$

For training points $\boldsymbol{X} = \{\boldsymbol{x}^{(n)}\}_{n=1}^N$ and test points $\boldsymbol{X}' = \{\boldsymbol{x}'^{(m)}\}_{m=1}^M$, we should set the the entries of the train-train $\boldsymbol{K} \in \mathbb{R}^{N \times N}$, train-test $\boldsymbol{K}' \in \mathbb{R}^{N \times M}$, and test-test $\boldsymbol{K}'' \in \mathbb{R}^{M \times M}$ kernels as

$$K_{n,n'} = \widetilde{k}^{(d(\boldsymbol{x}_n),d(\boldsymbol{x}_{n'}))}(\boldsymbol{x}_n, \boldsymbol{x}_{n'}), \tag{23}$$

$$K'_{n,m} = \widetilde{k}^{(d(\boldsymbol{x}_n),d(\boldsymbol{x}_m))}(\boldsymbol{x}_n, \boldsymbol{x}_m), \tag{24}$$

$$K''_{m,m'} = \widetilde{k}^{(d(\boldsymbol{x}_m),d(\boldsymbol{x}_{m'}))}(\boldsymbol{x}_m, \boldsymbol{x}_{m'}), \tag{25}$$

where

$$d(\boldsymbol{x}) = \begin{cases} 0 & \text{if the corresponding output for the input } \boldsymbol{x} \text{ is } f(\boldsymbol{x}), \\ d & \text{if the corresponding output for the input } \boldsymbol{x} \text{ is } \partial_{x_d} f(\boldsymbol{x}). \end{cases}$$

Eqs.(2) and (3) with the kernel matrices $\boldsymbol{K}, \boldsymbol{K}', \boldsymbol{K}''$ set as Eqs.(23)–(25) give the posterior GP for the corresponding test outputs.

For higher-order derivative outputs, we can define the kernels in exactly the same way as above, by applying the same derivative operators to the kernels as the ones applied to the outputs, i.e.,

$$\widetilde{k}(\boldsymbol{x}, \boldsymbol{x}') = \left[ \partial_{x_1}^{(r_1)} \cdots \partial_{x_D}^{(r_D)} \right] \left[ \partial_{x'_1}^{(r'_1)} \cdots \partial_{x'_D}^{(r'_D)} \right] k(\boldsymbol{x}, \boldsymbol{x}'),$$

if the corresponding outputs at $\boldsymbol{x}$ and $\boldsymbol{x}'$ are $\partial_{x_1}^{(r_1)} \cdots \partial_{x_D}^{(r_D)} f(\boldsymbol{x})$ and $\partial_{x'_1}^{(r'_1)} \cdots \partial_{x'_D}^{(r'_D)} f(\boldsymbol{x}')$, respectively, where $\partial_{x_d}^{(r)} \equiv \frac{\partial^r}{\partial x_d{}^r}$ denotes the $r$-th order derivative with respect to $x_d$.

## B  NAKANISHI-FUJI-TODO (NFT) ALGORITHM (NAKANISHI ET AL., 2020) AND BAYESIAN NFT

Let $\{\boldsymbol{e}_d\}_{d=1}^D$ be the standard basis. NFT is initialized with a random point $\widehat{\boldsymbol{x}}^0$ with a first observation $\widehat{y}^0 = f^*(\widehat{\boldsymbol{x}}^0) + \varepsilon_0$, and iterates the following procedure: for each iteration step $t$,

1. Select an axis $d \in \{1, \ldots, D\}$ sequentially and observe the objective $\boldsymbol{y} \in \mathbb{R}^{2V_d}$ at $2V_d$ points $\boldsymbol{X} = (\boldsymbol{x}_1, \ldots, \boldsymbol{x}_{2V_d}) = \{\widehat{\boldsymbol{x}}^{t-1} + \alpha_w \boldsymbol{e}_d\}_{w=1}^{2V_d} \in \mathbb{R}^{D \times 2V_d}$ along the axis $d$.[4] Here $\boldsymbol{\alpha} \in [0, 2\pi)^{2V_d}$ is such that $\alpha_w \neq 0$, $\alpha_{w'} \neq \alpha_w$, for all $w$ and $w' \neq w$.

2. Apply the 1D trigonometric polynomial regression $\widetilde{f}(\theta) = \widetilde{\boldsymbol{b}}^\top \boldsymbol{\psi}_1(\theta)$ to the $2V_d$ new observations $\boldsymbol{y}$, together with the previous best estimated score $\widehat{y}^{t-1}$, and analytically compute the new optimum $\widehat{\boldsymbol{x}}^t = \widehat{\boldsymbol{x}}^{t-1} + \widehat{\theta} \boldsymbol{e}_d$, where $\widehat{\theta} = \arg\min_\theta \widetilde{f}(\theta)$.

3. Update the best score by $\widehat{y}^t = \widetilde{f}(\widehat{\theta})$.

Note that if the observation noise is negligible, i.e., $y \approx f^*(\boldsymbol{x})$, each step of NFT reaches the global optimum in the 1D subspace along the chosen axis $d$ for any choice of $\boldsymbol{\alpha}$, and thus performs SMO

---

[4]With slight abuse of notation, we use the set notation to specify the column vectors of a matrix, i.e., $(\boldsymbol{x}_1, \ldots, \boldsymbol{x}_N) = \{\boldsymbol{x}_n\}_{n=1}^N$.

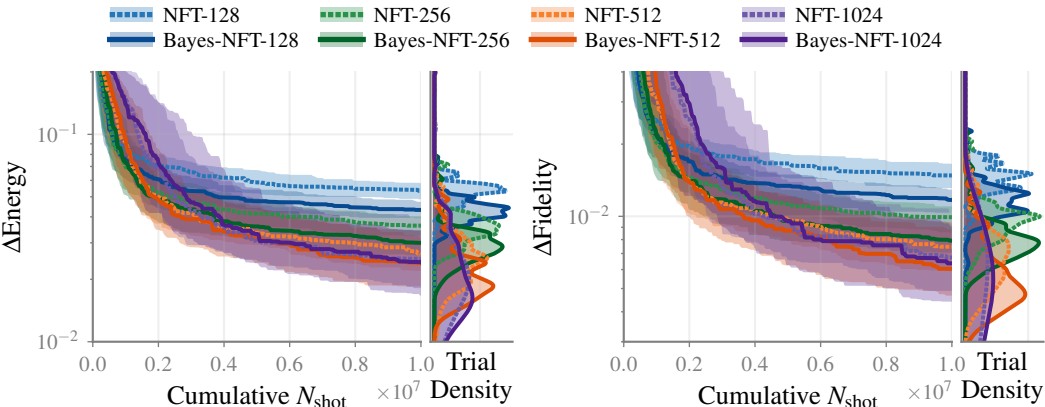

Figure 5: Comparison between NFT (Nakanishi et al., 2020) and Bayes-NFT for the Ising Hamiltonian with a $(Q = 5)$-qubits $(L = 3)$-layers quantum circuit. The energy (left) and fidelity (right), in the forms of Eqs.(21) and (22), respectively, are plotted as functions of the cumulative $N_{\text{shots}}$, i.e., the total number of measurement shots. The number of shots per observation is set to $N_{\text{shots}} = 128$ (blue), 256 (green), 512 (orange), and 1024 (purple).

exactly. Otherwise, errors can be accumulated in the best score $\widehat{y}^t$, and therefore an additional measurement may need to be performed at $\widehat{x}^t$ after a certain iteration interval.

Bayesian NFT (Bayes-NFT) performs the 1D trigonometric polynomial regression and optimization in Step 2 with GP with the VQE kernel (13), where all previous observations are used for training. Using previous observations allows prediction with smaller uncertainty and thus more accurate subspace optimization. Figure 5 compares the original NFT and Bayesian NFT on the Ising Hamiltonian with a $(Q = 5)$-qubits $(L = 3)$-layers quantum circuit with different number of shots per observation. We observe that using GP generally accelerates the optimization process.

## C   PROOFS

Here, we give proofs of theorems in Section 3, and numerically validate them.

### C.1   PROOF OF THEOREM 3.1

We start from a more general theorem than Theorem 3.1, which is proven in Appendix C.3.

**Theorem C.1.** *Assume that, for any given point $\widehat{x} \in [0, 2\pi)^D$, we have observations $y = (y_0, \ldots, y_{2V_d-1})^\top \in \mathbb{R}^{2V_d}$ at $2V_d$ equidistant training points $X = (x_0, \ldots, x_{2V_d-1}) \in \mathbb{R}^{D \times 2V_d}$ for $x_w = \widehat{x} + \frac{2w+1}{2V_d}\pi e_d$ with homoschedastic noise $\sigma = \sigma^2 \cdot \mathbf{1}_{2V_d} \in \mathbb{R}^{2V_d}$. Then, the mean and variance of the derivative $\partial_d f(x')$ prediction at $x' = \widehat{x} + \alpha' e_d$ for any $d = 1, \ldots, D$ and $\alpha' \in [0, 2\pi)$ are given as*

$$\widetilde{\mu}_{[X,y,\sigma]}^{(d)}(x') = \frac{1}{(\gamma^2+2V_d)\sigma^2/\sigma_0^2+2V_d} \sum_{w=0}^{2V_d-1} (-1)^w y_w$$
$$\cdot \left( \frac{\cos(V_d\alpha')}{2\sin^2\left(\frac{(2w+1)\pi}{4V_d}-\alpha'/2\right)} + \frac{V_d\sin\left(\frac{(2w+1)\pi}{4V_d}-(V_d+1/2)\alpha'\right)}{\sin\left(\frac{(2w+1)\pi}{4V_d}-\alpha'/2\right)} - \frac{4V_d^2\cos V_d\alpha'}{(\gamma^2+2V_d)\sigma^2/\sigma_0^2+4V_d} \right), \quad (26)$$

$$\widetilde{s}_{[X,\sigma]}^{(d)}(x',x') = \sigma^2 \left( \frac{V_d(V_d+1)(2V_d+1)}{3((\gamma^2+2V_d)\sigma^2/\sigma_0^2+2V_d)} - \frac{4V_d^3\cos(2V_d\alpha')}{((\gamma^2+2V_d)\sigma^2/\sigma_0^2+2V_d)((\gamma^2+2V_d)\sigma^2/\sigma_0^2+4V_d)} \right)$$
$$- \sigma_0^2 \frac{8V_d^4(\cos(2V_d\alpha')-1)}{(\gamma^2+2V_d)((\gamma^2+2V_d)\sigma^2/\sigma_0^2+2V_d)((\gamma^2+2V_d)\sigma^2/\sigma_0^2+4V_d)}. \quad (27)$$

Regardless of the observations, the predictive uncertainty (27) is periodic with respective to $\alpha'$ with the period of $\pi/V_d$. We can easily get the following corollaries.

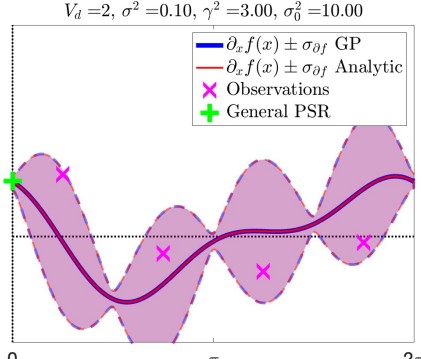 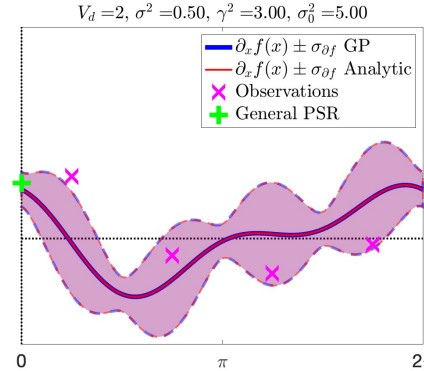

Figure 6: Numerical validation of Theorem C.1 under two parameter settings (see above each panel). Given the $2V_d$ equidistant observations (magenta crosses), the derivative GP prediction (blue curve) with uncertainty (blue dashed curves) is compared to their analytic forms (26) and (27), i.e., the mean function (red curve) and the variance function (red dashed curves), respectively. We observe that our theory perfectly matches the numerical computation. The green cross shows the prediction by the general PSR (10), which almost coincides with Bayesian PSR prediction when $\sigma^2/\sigma_0^2 = 0.01$ (left panel), while a significant difference is observed when $\sigma^2/\sigma_0^2 = 0.1$ (right panel).

**Corollary C.2.** *For the test point at $\boldsymbol{x}' = \widehat{\boldsymbol{x}}$, i.e., $\alpha' = 0$, the mean of the derivative GP prediction is*

$$\widetilde{\mu}_{[\boldsymbol{X},\boldsymbol{y},\boldsymbol{\sigma}]}^{(d)}(\boldsymbol{x}') = \frac{\sum_{w=0}^{2V_d-1}(-1)^w y_w \left(\frac{1}{2\sin^2\left(\frac{(2w+1)\pi}{4V_d}\right)} + \frac{V_d(\gamma^2+2V_d)\sigma^2/\sigma_0^2}{(\gamma^2+2V_d)\sigma^2/\sigma_0^2+4V_d}\right)}{(\gamma^2+2V_d)\sigma^2/\sigma_0^2+2V_d}, \tag{28}$$

**Corollary C.3.** *For the test point at $\boldsymbol{x}' = \widehat{\boldsymbol{x}} + \alpha'\boldsymbol{e}_d, \forall \alpha' = 0, \pi/V_d, 2\pi/V_d, \ldots, (2V_d-1)\pi/V_d$, the variance of the derivative GP prediction is*

$$\widetilde{s}_{[\boldsymbol{X},\boldsymbol{\sigma}]}^{(d)}(\boldsymbol{x}',\boldsymbol{x}') = \sigma^2 \left(\frac{V_d(V_d+1)(2V_d+1)(\gamma^2+2V_d)\sigma^2/\sigma_0^2+4V_d^2(2V_d^2+1)}{3((\gamma^2+2V_d)\sigma^2/\sigma_0^2+2V_d)((\gamma^2+2V_d)\sigma^2/\sigma_0^2+4V_d)}\right). \tag{29}$$

Ignoring high order terms with respect to $\sigma^2/\sigma_0^2$ in Eqs.(28) and (29) gives Theorem 3.1. $\qquad\square$

Figure 6 shows numerical validation of Theorem C.1, where the derivative GP prediction (blue curve) with uncertainty (blue dashed curves) is compared to their analytic forms, i.e., the mean function (26) (red curve) and the variance function (27) (red dashed curves), respectively, under two settings of noise and kernel parameters. We observe that our theory perfectly matches the numerical computation. When $\sigma^2/\sigma_0^2 = 0.01$ (left panel), the regularization is small enough and the Bayesian PSR prediction (red curve) almost coincides with the general PSR prediction (green cross). On the other hand, when $\sigma^2/\sigma_0^2 = 0.1$ (right panel), the Bayesian PSR prediction (red) does not match the general PSR prediction (green cross), because of the regularization.

In addition, we can also derive an upper bound of Eq.(29), which is useful for setting the grid search range in the SGD-GradCoRe algorithm described in Appendix D.

**Corollary C.4.** *For the test point at $\boldsymbol{x}' = \widehat{\boldsymbol{x}} + \alpha'\boldsymbol{e}_d, \forall \alpha' = 0, \pi/V_d, 2\pi/V_d, \ldots, (2V_d-1)\pi/V_d$, the variance of the derivative GP prediction is upper bounded as*

$$\widetilde{s}_{[\boldsymbol{X},\boldsymbol{\sigma}]}^{(d)}(\boldsymbol{x}',\boldsymbol{x}') < \sigma^2 \left(\frac{2V_d^2+1}{6}\right). \tag{30}$$

*Proof.* Let $\xi = (\gamma^2 + 2V_d)\sigma^2/\sigma_0^2 > 0$. Then, Eq.(29) can be written as

$$\widetilde{s}_{[\boldsymbol{X},\boldsymbol{\sigma}]}^{(d)}(\boldsymbol{x}',\boldsymbol{x}') = \sigma^2 \left(\frac{V_d(V_d+1)(2V_d+1)\xi+4V_d^2(2V_d^2+1)}{3(\xi+2V_d)(\xi+4V_d)}\right), \tag{31}$$

and its derivative with respect to $\xi$ is given as

$$\frac{\partial \widetilde{s}_{[\boldsymbol{X},\boldsymbol{\sigma}]}^{(d)}(\boldsymbol{x}',\boldsymbol{x}')}{\partial \xi} = -\sigma^2 \left(\frac{V_d(V_d+1)(2V_d+1)\xi^2+8V_d^2(2V_d^2+1)\xi+8V_d^3(4V_d^2-3V_d+2)}{3(\xi+2V_d)^2(\xi+4V_d)^2}\right). \tag{32}$$

Apparently, Eq.(32) is negative for any $V_d \geq 1$, and therefore, the predictive variance $\widetilde{s}_{[\boldsymbol{X},\boldsymbol{\sigma}]}^{(d)}(\boldsymbol{x}',\boldsymbol{x}')$ is monotonically decreasing with respect to $\xi > 0$. Taking the limit $\xi \to 0$, Eq.(31) reduces to the bound (30). $\qquad\square$

Note that the monotonicity of the predictive variance with respect to $\xi$, and hence with respect to $\sigma_0^2$ (when $\gamma^2$ and $\sigma^2$ are fixed), matches the intuition that the posterior uncertainty is smaller when the prior variance is smaller.

## C.2 MATHEMATICAL PREPARATIONS

Before proving Theorem C.1, we give some mathematical identities on the trigonometric functions.

### C.2.1 ROOT OF UNITY

For a natural number $N \in \{1, 2, \ldots\}$, let us define a root of unity $\rho_N = e^{2\pi i/N}$ such that $\rho_N^N = 1$. Then, the following hold:

$$\sum_{n=0}^{N-1} \rho_N^{nk} = \frac{1-\rho_N^{kN}}{1-\rho_N^k} = 0 \qquad \text{for} \qquad k = 1, \ldots, N-1, \qquad (33)$$

$$\sum_{n=0}^{N-1} \rho_N^{(n+\phi)k} = \rho_N^{k\phi} \sum_{n=0}^{N-1} \rho_N^{nk} = \rho_N^{k\phi} \frac{1-\rho_N^{kN}}{1-\rho_N^k} = 0 \qquad \text{for} \qquad k = 1, \ldots, N-1, \qquad (34)$$

It also holds for even $N$ that

$$\sum_{n=0}^{N-1} \rho_N^{(n+1/2)k+nN/2} = \rho_N^{k/2} \sum_{n=0}^{N-1} \rho_N^{n(k+N/2)} = \rho_N^{k/2} \frac{1-\rho_N^{(k+N/2)N}}{1-\rho_N^{(k+N/2)}} = 0 \qquad (35)$$

for $k = 1, \ldots, N/2 - 1$.

### C.2.2 PROPERTIES OF DIRICHLET KERNEL

The summation in the Dirichlet kernel can be analytically performed as

$$\begin{aligned}
1 + 2\sum_{n=1}^{N} \cos(nx) = 1 + 2\sum_{n=1}^{N} \frac{e^{inx}+e^{-inx}}{2} &= \sum_{n=-N}^{N} e^{inx} \\
&= e^{-iNx} \frac{1-e^{i(2N+1)x}}{1-e^{ix}} \\
&= \frac{e^{-i(N+1/2)x}-e^{i(N+1/2)x}}{e^{-ix/2}-e^{ix/2}} \\
&= \frac{\sin((N+1/2)x)}{\sin(x/2)}.
\end{aligned} \qquad (36)$$

Therefore, it also holds that

$$\begin{aligned}
2\sum_{n=1}^{N} n\sin(nx) &= -\sum_{v=1}^{V_d} \frac{\partial}{\partial x}\left(1/V_d + 2\cos(nx)\right) \\
&= -\frac{\partial}{\partial x}\left(1 + 2\sum_{v=1}^{V_d} \cos(nx)\right) \\
&= -\frac{(N+1/2)\cos((N+1/2)x)\sin(x/2)-\frac{1}{2}\sin((N+1/2)x)\cos(x/2)}{\sin^2(x/2)} \\
&= -\frac{N\cos((N+1/2)x)\sin(x/2)-\frac{1}{2}\sin(Nx)}{\sin^2(x/2)} \\
&= \frac{\sin(Nx)}{2\sin^2(x/2)} - \frac{N\cos((N+1/2)x)}{\sin(x/2)}.
\end{aligned} \qquad (37)$$

## C.3 PROOF OF THEOREM C.1

For derivative predictions $\partial_d f(\boldsymbol{x}'), \partial_d f(\boldsymbol{x}'')$, the test kernels should be modified as Eqs.(4) and (5). For the VQE kernel (13), they are

$$\begin{aligned}
\widetilde{k}(\boldsymbol{x},\boldsymbol{x}') &= \partial_{x'_d} k(\boldsymbol{x},\boldsymbol{x}') \\
&= \sigma_0^2 \left(\frac{2\sum_{v=1}^{V_d} v\sin(v(x_d-x'_d))}{\gamma^2+2V_d}\right) \prod_{d'\neq d}\left(\frac{\gamma^2+2\sum_{v=1}^{V_{d'}} \cos(v(x_{d'}-x'_{d'}))}{\gamma^2+2V_{d'}}\right),
\end{aligned} \qquad (38)$$

$$\widetilde{k}(\boldsymbol{x}', \boldsymbol{x}'') = \partial_{x'_d}\partial_{x''_d}k(\boldsymbol{x}', \boldsymbol{x}'')$$

$$= \sigma_0^2 \left( \frac{2\sum_{v=1}^{V_d} v^2 \cos\left(v(x'_d - x''_d)\right)}{\gamma^2 + 2V_d} \right) \prod_{d' \neq d} \left( \frac{\gamma^2 + 2\sum_{v=1}^{V_{d'}} \cos\left(v(x'_{d'} - x''_{d'})\right)}{\gamma^2 + 2V_{d'}} \right). \quad (39)$$

The training kernel matrix for $\{\boldsymbol{x}_w = \widehat{\boldsymbol{x}} + \frac{2w+1}{2V_d}\pi\boldsymbol{e}_d\}_{w=0}^{2V_d-1}$ is Toeplitz as

$$\boldsymbol{K} = \sigma_0^2 \begin{pmatrix} \tau_0 & \tau_1 & \tau_2 & \cdots & \tau_{2V_d-2} & \tau_{2V_d-1} \\ \tau_1 & \tau_0 & \tau_1 & & & \\ \tau_2 & \tau_1 & \tau_0 & & & \vdots \\ \vdots & & & \ddots & & \\ \tau_{2V_d-2} & & & & \tau_0 & \tau_1 \\ \tau_{2V_d-1} & & \cdots & & \tau_1 & \tau_0 \end{pmatrix} \in \mathbb{R}^{2V_d \times 2V_d},$$

where

$$\tau_w = \frac{\gamma^2 + 2\sum_{v=1}^{V_d} \cos\left(\frac{vw}{2V_d}2\pi\right)}{\gamma^2 + 2V_d}. \quad (40)$$

For a test point at $\boldsymbol{x}' = \hat{\boldsymbol{x}} + \alpha'\boldsymbol{e}_d$, the test kernel components are

$$\widetilde{\boldsymbol{k}}' = \sigma_0^2 \begin{pmatrix} \kappa_0 \\ \kappa_1 \\ \vdots \\ \kappa_{2V_d-1} \end{pmatrix},$$

$$\widetilde{k}'' = \sigma_0^2,$$

where

$$\kappa_w = \frac{2\sum_{v=1}^{V_d} v\sin\left(v\left(\frac{2w+1}{2V_d}\pi - \alpha'\right)\right)}{\gamma^2 + 2V_d}. \quad (41)$$

The first identity (33) for the root of unity implies that

$$\sum_{v=0}^{2V_d-1} e^{vw\frac{2\pi i}{2V_d}} = 0 \qquad \text{for} \qquad w = 1, \dots, 2V_d - 1,$$

and therefore

$$\sum_{v=0}^{2V_d-1} \cos\left(vw\frac{2\pi}{2V_d}\right) = \begin{cases} 2V_d & \text{for} & w = 0, 2V_d, \\ 0 & \text{for} & w = 1, \dots, 2V_d - 1, \end{cases} \quad (42)$$

$$\sum_{v=0}^{2V_d-1} \sin\left(vw\frac{2\pi}{2V_d}\right) = 0 \qquad \text{for} \qquad w = 0, \dots, 2V_d. \quad (43)$$

The second identity (34) for the root of unity gives

$$\sum_{v=0}^{2V_d-1} e^{(v+1/2)w\frac{2\pi i}{2V_d}} = \sum_{v=0}^{2V_d-1} e^{(2v+1)w\frac{\pi i}{2V_d}} = 0 \qquad \text{for} \qquad w = 1, \dots, 2V_d - 1,$$

and therefore

$$\sum_{v=0}^{2V_d-1} \cos\left((2v+1)w\frac{\pi}{2V_d}\right) = \begin{cases} 2V_d & \text{for} & w = 0, \\ -2V_d & \text{for} & w = 2V_d, \\ 0 & \text{for} & w = 1, \dots, 2V_d - 1, \end{cases} \quad (44)$$

$$\sum_{v=0}^{2V_d-1} \sin\left((2v+1)w\frac{\pi}{2V_d}\right) = 0 \qquad \text{for} \qquad w = 0, \dots, 2V_d. \quad (45)$$

The third identity (35) for the root of unity gives

$$\sum_{v=0}^{2V_d-1} e^{((v+1/2)w+vV_d)\frac{2\pi i}{2V_d}} = \sum_{v=0}^{2V_d-1} e^{v\pi i} e^{(2v+1)w\frac{\pi i}{2V_d}} = \sum_{v=0}^{2V_d-1} (-1)^v e^{(2v+1)w\frac{\pi i}{2V_d}} = 0$$

for $w = 1, \ldots, V_d - 1$, and therefore

$$\sum_{v=0}^{2V_d-1} (-1)^v \cos\left((2v+1)w\frac{\pi}{2V_d}\right) = 0 \qquad \text{for} \qquad w = 0, \ldots, V_d, \tag{46}$$

$$\sum_{v=0}^{2V_d-1} (-1)^v \sin\left((2v+1)w\frac{\pi}{2V_d}\right) = \begin{cases} 2V_d & \text{for} \qquad w = V_d, \\ 0 & \text{for} \qquad w = 0, \ldots, V_d - 1. \end{cases} \tag{47}$$

From the symmetry of the trigonometric functions, it holds that

$$\sum_{v=1}^{V_d} \cos\left(vw\frac{2\pi}{2V_d}\right) = \begin{cases} -1 & \text{for} \qquad w = 1, 3, 5, \ldots, 2V_d - 1, \\ 0 & \text{for} \qquad w = 2, 4, 6, \ldots, 2V_d, \end{cases} \tag{48}$$

$$\sum_{v=1}^{V_d} \sin\left(vw\frac{2\pi}{2V_d}\right) = -\sum_{v=V_d+1}^{2V_d} \sin\left(vw\frac{2\pi}{2V_d}\right). \tag{49}$$

Note that the factor $-1$ in the odd $w$ case in Eq. (48) is because the summand is $-1$ for $v = V_d$, while the summands for the other $v$ are canceled each other.

By using Eq. (48), Eq. (40) can be written as

$$\tau_w = \frac{\gamma^2 + 2\sum_{v=1}^{V_d} \cos\left(\frac{vw}{2V_d} 2\pi\right)}{\gamma^2 + 2V_d} = \begin{cases} 1 & \text{for} \qquad w = 0, \\ \frac{\gamma^2 - 2}{\gamma^2 + 2V_d} & \text{for} \qquad w = 1, 3, 5, \ldots, 2V_d - 1, \\ \frac{\gamma^2}{\gamma^2 + 2V_d} & \text{for} \qquad w = 2, 4, 6, \ldots, 2V_d - 2, \end{cases}$$

and therefore

$$\begin{aligned}
\boldsymbol{K} &= \frac{\sigma_0^2}{\gamma^2 + 2V_d} \left(2V_d \boldsymbol{I}_{2V_d} + (\gamma^2 - 1)\boldsymbol{1}\boldsymbol{1}^\top + \boldsymbol{c}\boldsymbol{c}^\top\right) \\
&= \frac{\sigma_0^2}{\gamma^2 + 2V_d} \left(2V_d \boldsymbol{I}_{2V_d} + (\boldsymbol{1} \quad \boldsymbol{c})\begin{pmatrix} \gamma^2 - 1 & 0 \\ 0 & 1 \end{pmatrix}(\boldsymbol{1} \quad \boldsymbol{c})^\top\right),
\end{aligned} \tag{50}$$

where

$$\boldsymbol{c} = \begin{pmatrix} 1 \\ -1 \\ 1 \\ -1 \\ \vdots \\ 1 \\ -1 \end{pmatrix} \in \mathbb{R}^{2V_d}.$$

With the training kernel expression (50), the matrix inversion lemma gives

$$\begin{aligned}
\left(\boldsymbol{K} + \sigma^2 \boldsymbol{I}_{2V_d}\right)^{-1} &= \frac{\gamma^2 + 2V_d}{\sigma_0^2} \left((\gamma^2 + 2V_d)(\sigma^2/\sigma_0^2 + 2V_d)\,\boldsymbol{I}_{2V_d} + (\boldsymbol{1} \quad \boldsymbol{c})\begin{pmatrix} \gamma^2 - 1 & 0 \\ 0 & 1 \end{pmatrix}(\boldsymbol{1} \quad \boldsymbol{c})^\top\right)^{-1} \\
&= \frac{\gamma^2 + 2V_d}{\sigma_0^2} \frac{1}{(\gamma^2 + 2V_d)\sigma^2/\sigma_0^2 + 2V_d} \\
&\quad \left(\boldsymbol{I}_{2V_d} + \frac{1}{(\gamma^2 + 2V_d)\sigma^2/\sigma_0^2 + 2V_d}(\boldsymbol{1} \quad \boldsymbol{c})\begin{pmatrix} \gamma^2 - 1 & 0 \\ 0 & 1 \end{pmatrix}(\boldsymbol{1} \quad \boldsymbol{c})^\top\right)^{-1} \\
&= \frac{\gamma^2 + 2V_d}{\sigma_0^2} \frac{1}{(\gamma^2 + 2V_d)\sigma^2/\sigma_0^2 + 2V_d} \\
&\quad \left\{\boldsymbol{I}_{2V_d} - \frac{1}{(\gamma^2 + 2V_d)\sigma^2/\sigma_0^2 + 2V_d}(\boldsymbol{1} \quad \boldsymbol{c})\begin{pmatrix} \gamma^2 - 1 & 0 \\ 0 & 1 \end{pmatrix}\right.
\end{aligned}$$

$$\left( \boldsymbol{I}_2 + (\boldsymbol{1} \quad \boldsymbol{c})^\top \tfrac{1}{(\gamma^2+2V_d)\sigma^2/\sigma_0^2+2V_d} (\boldsymbol{1} \quad \boldsymbol{c}) \begin{pmatrix} \gamma^2-1 & 0 \\ 0 & 1 \end{pmatrix} \right)^{-1} (\boldsymbol{1} \quad \boldsymbol{c})^\top \Bigg\}$$

$$= \tfrac{\gamma^2+2V_d}{\sigma_0^2} \tfrac{1}{(\gamma^2+2V_d)\sigma^2/\sigma_0^2+2V_d}$$

$$\Bigg\{ \boldsymbol{I}_{2V_d} - \tfrac{1}{(\gamma^2+2V_d)\sigma^2/\sigma_0^2+2V_d} (\boldsymbol{1} \quad \boldsymbol{c}) \begin{pmatrix} \gamma^2-1 & 0 \\ 0 & 1 \end{pmatrix}$$

$$\left( \boldsymbol{I}_2 + \tfrac{1}{(\gamma^2+2V_d)\sigma^2/\sigma_0^2+2V_d} \begin{pmatrix} 2V_d(\gamma^2-1) & 0 \\ 0 & 2V_d \end{pmatrix} \right)^{-1} (\boldsymbol{1} \quad \boldsymbol{c})^\top \Bigg\}$$

$$= \tfrac{\gamma^2+2V_d}{\sigma_0^2} \tfrac{1}{(\gamma^2+2V_d)\sigma^2/\sigma_0^2+2V_d}$$

$$\Bigg\{ \boldsymbol{I}_{2V_d} - (\boldsymbol{1} \quad \boldsymbol{c}) \begin{pmatrix} \gamma^2-1 & 0 \\ 0 & 1 \end{pmatrix}$$

$$\left( \begin{pmatrix} (\gamma^2+2V_d)\sigma^2/\sigma_0^2+2V_d\gamma^2 & 0 \\ 0 & (\gamma^2+2V_d)\sigma^2/\sigma_0^2+4V_d \end{pmatrix} \right)^{-1} (\boldsymbol{1} \quad \boldsymbol{c})^\top \Bigg\}$$

$$= \tfrac{1}{\sigma_0^2} a (\boldsymbol{I}_{2V_d} + b\boldsymbol{1}\boldsymbol{1}^\top + c\boldsymbol{c}\boldsymbol{c}^\top),$$

where

$$a = \frac{\gamma^2 + 2V_d}{(\gamma^2 + 2V_d)\sigma^2/\sigma_0^2 + 2V_d},$$

$$b = -\frac{\gamma^2 - 1}{(\gamma^2 + 2V_d)\sigma^2/\sigma_0^2 + 2V_d\gamma^2},$$

$$c = -\frac{1}{(\gamma^2 + 2V_d)\sigma^2/\sigma_0^2 + 4V_d}.$$

For the test kernels

$$\widetilde{\boldsymbol{k}}' = \sigma_0^2 \begin{pmatrix} \kappa_0 \\ \kappa_1 \\ \vdots \\ \kappa_{2V_d-1} \end{pmatrix},$$

$$\widetilde{\boldsymbol{k}}'' = \sigma_0^2 \left( \frac{2\sum_{v=1}^{V_{d'}} v^2}{\gamma^2 + 2V_d} \right) = \frac{\sigma_0^2 V_d(V_d+1)(2V_d+1)}{3(\gamma^2 + 2V_d)},$$

with

$$\kappa_w = \frac{2\sum_{v=1}^{V_d} v \sin\left( v\left( \frac{(2w+1)\pi}{2V_d} - \alpha' \right) \right)}{\gamma^2 + 2V_d},$$

(51)

we have

$$\|\widetilde{\boldsymbol{k}}'\|^2 = \sigma_0^4 \sum_{w=0}^{2V_d-1} \left( \frac{2\sum_{v=1}^{V_d} v \sin\left( v\left( \frac{(2w+1)\pi}{2V_d} - \alpha' \right) \right)}{\gamma^2 + 2V_d} \right)^2$$

$$= \frac{\sigma_0^4}{(\gamma^2+2V_d)^2} \sum_{w=0}^{2V_d-1} \left\{ 4\sum_{v=1}^{V_d}\sum_{v'=1}^{V_d} vv' \sin\left( v\left( \frac{(2w+1)\pi}{2V_d} - \alpha' \right) \right) \sin\left( v'\left( \frac{(2w+1)\pi}{2V_d} - \alpha' \right) \right) \right\}$$

$$= \frac{\sigma_0^4}{(\gamma^2 + 2V_d)^2} \sum_{w=0}^{2V_d-1} \Bigg\{ 2\sum_{v=1}^{V_d}\sum_{v'=1}^{V_d} vv'$$

$$\cdot \left( \cos\left( (v-v')\left( \frac{(2w+1)\pi}{2V_d} - \alpha' \right) \right) - \cos\left( (v+v')\left( \frac{(2w+1)\pi}{2V_d} - \alpha' \right) \right) \right) \Bigg\}$$

$$= \frac{\sigma_0^4}{(\gamma^2 + 2V_d)^2} \left\{ 2 \sum_{v=1}^{V_d} \sum_{v'=1}^{V_d} vv' \sum_{w=0}^{2V_d-1} \right.$$

$$\left. \left( \cos\left( (v - v')\left( \frac{(2w+1)\pi}{2V_d} - \alpha' \right) \right) - \cos\left( (v + v')\left( \frac{(2w+1)\pi}{2V_d} - \alpha' \right) \right) \right) \right\}$$

$$= \frac{\sigma_0^4}{(\gamma^2 + 2V_d)^2} \left\{ 2 \sum_{v=1}^{V_d} \sum_{v'=1}^{V_d} vv' \sum_{w=0}^{2V_d-1} \right.$$

$$\left( \cos\frac{(2w+1)(v-v')\pi}{2V_d} \cos\left( (v-v')\alpha' \right) + \sin\frac{(2w+1)(v-v')\pi}{2V_d} \sin\left( (v-v')\alpha' \right) \right.$$

$$\left. \left. - \cos\frac{(2w+1)(v+v')\pi}{2V_d} \cos\left( (v+v')\alpha' \right) - \sin\frac{(2w+1)(v+v')\pi}{2V_d} \sin\left( (v+v')\alpha' \right) \right) \right\}$$

(52)

$$= \frac{\sigma_0^4}{(\gamma^2 + 2V_d)^2} \left\{ 2 \sum_{v=1}^{V_d} \sum_{v'=1}^{V_d} vv' \sum_{w=0}^{2V_d-1} \left( \cos\frac{(2w+1)(v-v')\pi}{2V_d} \cos\left( (v-v')\alpha' \right) \right. \right.$$

$$\left. \left. - \cos\frac{(2w+1)(v+v')\pi}{2V_d} \cos\left( (v+v')\alpha' \right) \right) \right\}$$

(53)

$$= \frac{\sigma_0^4}{(\gamma^2 + 2V_d)^2} 2(2V_d) \left( \left( \sum_{v=1}^{V_d} v^2 \right) + V_d^2 \cos\left( 2V_d\alpha' \right) \right)$$

(54)

$$= \frac{\sigma_0^4}{(\gamma^2 + 2V_d)^2} 2(2V_d) \left( \frac{V_d(V_d+1)(2V_d+1)}{6} + V_d^2 \cos\left( 2V_d\alpha' \right) \right)$$

$$= \sigma_0^4 \frac{4V_d^2}{(\gamma^2 + 2V_d)^2} \left( \frac{(V_d+1)(2V_d+1)}{6} + V_d \cos\left( 2V_d\alpha' \right) \right).$$

Here we used Eqs.(44) and (45) to obtain Eqs.(53) and (54) from Eq. (52).

We also have

$$\|\widetilde{k}'\|_1 = \widetilde{k}'^\top \mathbf{1}_{2V_d} = \sigma_0^2 \sum_{w=0}^{2V_d-1} \frac{2\sum_{v=1}^{V_d} v \sin\left( v\left( \frac{(2w+1)\pi}{2V_d} - \alpha' \right) \right)}{\gamma^2 + 2V_d}$$

$$= \sigma_0^2 \frac{2\sum_{v=1}^{V_d} v \sum_{w=0}^{2V_d-1} \sin\left( v\left( \frac{(2w+1)\pi}{2V_d} - \alpha' \right) \right)}{\gamma^2 + 2V_d}$$

$$= \sigma_0^2 \frac{2\sum_{v=1}^{V_d} v \sum_{w=0}^{2V_d-1} \left( \sin\frac{(2w+1)v\pi}{2V_d} \cos v\alpha' - \cos\frac{(2w+1)v\pi}{2V_d} \sin v\alpha' \right)}{\gamma^2 + 2V_d}$$

$$= 0,$$

and

$$\widetilde{k}'^\top c = \sigma_0^2 \sum_{w=0}^{2V_d-1} (-1)^w \frac{2\sum_{v=1}^{V_d} v \sin\left( v\left( \frac{(2w+1)\pi}{2V_d} - \alpha' \right) \right)}{\gamma^2 + 2V_d}$$

$$= \sigma_0^2 \frac{2\sum_{v=1}^{V_d} v \sum_{w=0}^{2V_d-1} (-1)^w \sin\left( v\left( \frac{(2w+1)\pi}{2V_d} - \alpha' \right) \right)}{\gamma^2 + 2V_d}$$

$$= \sigma_0^2 \frac{2\sum_{v=1}^{V_d} v \sum_{w=0}^{2V_d-1} (-1)^w \left( \sin\frac{(2w+1)v\pi}{2V_d} \cos v\alpha' - \cos\frac{(2w+1)v\pi}{2V_d} \sin v\alpha' \right)}{\gamma^2 + 2V_d}$$

$$= \sigma_0^2 \frac{2V_d 2V_d \cos V_d\alpha'}{\gamma^2 + 2V_d}$$

$$= \sigma_0^2 \frac{4V_d^2 \cos V_d \alpha'}{\gamma^2 + 2V_d}.$$

Here, we used Eqs.(46) and (47) in the second last equation. Therefore, the mean of the derivative is

$$
\begin{aligned}
\widetilde{\mu}^{(d)}_{[\boldsymbol{X},\boldsymbol{y},\boldsymbol{\sigma}]}(\boldsymbol{x}') &= \widetilde{\boldsymbol{k}}'^{\top} \left(\boldsymbol{K} + \sigma^2 \boldsymbol{I}_{2V_d}\right)^{-1} \boldsymbol{y} \\
&= \widetilde{\boldsymbol{k}}'^{\top} \frac{a}{\sigma_0^2} \left(\boldsymbol{I}_{2V_d} + b\boldsymbol{1}_{2V_d}\boldsymbol{1}_{2V_d}^{\top} + c\boldsymbol{c}\boldsymbol{c}^{\top}\right) \boldsymbol{y} \\
&= \frac{a}{\sigma_0^2} \left(\widetilde{\boldsymbol{k}}'^{\top} \boldsymbol{y} + b\widetilde{\boldsymbol{k}}'^{\top} \boldsymbol{1}_{2V_d}\boldsymbol{1}_{2V_d}^{\top}\boldsymbol{y} + c\widetilde{\boldsymbol{k}}'^{\top} \boldsymbol{c}\boldsymbol{c}^{\top}\boldsymbol{y}\right) \\
&= a\left(\sum_{w=0}^{2V_d-1} y_w \frac{2\sum_{v=1}^{V_d} v \sin\left(v\left(\frac{(2w+1)\pi}{2V_d} - \alpha'\right)\right)}{\gamma^2 + 2V_d} + c\frac{4V_d^2 \cos V_d\alpha'}{\gamma^2+2V_d} \sum_{w=0}^{2V_d-1} (-1)^w y_w\right). \\
&= a\left(\sum_{w=0}^{2V_d-1} y_w \left(\frac{2\sum_{v=1}^{V_d} v \sin\left(v\left(\frac{(2w+1)\pi}{2V_d} - \alpha'\right)\right)}{\gamma^2 + 2V_d} + c\frac{4V_d^2 (-1)^w}{\gamma^2+2V_d} \cos V_d\alpha'\right)\right) \\
&= \frac{a}{\gamma^2+2V_d}\left(\sum_{w=0}^{2V_d-1} y_w \left(\left\{2\sum_{v=1}^{V_d} v \sin\left(v\left(\frac{(2w+1)\pi}{2V_d} - \alpha'\right)\right)\right\} + 4cV_d^2 (-1)^w \cos V_d\alpha'\right)\right) \\
&= \frac{\sum_{w=0}^{2V_d-1} y_w \left(\left\{2\sum_{v=1}^{V_d} v \sin\left(v\left(\frac{(2w+1)\pi}{2V_d} - \alpha'\right)\right)\right\} - \frac{4V_d^2(-1)^w \cos V_d\alpha'}{(\gamma^2+2V_d)\sigma^2/\sigma_0^2 + 4V_d}\right)}{(\gamma^2+2V_d)\sigma^2/\sigma_0^2 + 2V_d}.
\end{aligned}
\tag{55}
$$

Eq. (37) implies that, for $w = 0, 1, \ldots, 2V_d - 1$, it holds that

$$
\begin{aligned}
2\sum_{v=1}^{V_d} v\sin\left(v\left(\frac{(2w+1)\pi}{2V_d} - \alpha'\right)\right) &= \frac{\sin\left(V_d\left(\frac{(2w+1)\pi}{2V_d} - \alpha'\right)\right)}{2\sin^2\left(\left(\frac{(2w+1)\pi}{2V_d} - \alpha'\right)/2\right)} - \frac{V_d\cos\left((V_d+1/2)\left(\frac{(2w+1)\pi}{2V_d} - \alpha'\right)\right)}{\sin\left(\left(\frac{(2w+1)\pi}{2V_d} - \alpha'\right)/2\right)} \\
&= \frac{\sin\left(\frac{(2w+1)\pi}{2} - V_d\alpha'\right)}{2\sin^2\left(\left(\frac{(2w+1)\pi}{2V_d} - \alpha'\right)/2\right)} - \frac{V_d\cos\left(\frac{(2w+1)\pi}{2} + \frac{(2w+1)\pi}{4V_d} - (V_d+1/2)\alpha'\right)}{\sin\left(\left(\frac{(2w+1)\pi}{2V_d} - \alpha'\right)/2\right)} \\
&= \frac{\sin\left((-1)^w\frac{\pi}{2} - V_d\alpha'\right)}{2\sin^2\left(\left(\frac{(2w+1)\pi}{2V_d} - \alpha'\right)/2\right)} - \frac{V_d\cos\left((-1)^w\frac{\pi}{2} + \frac{(2w+1)\pi}{4V_d} - (V_d+1/2)\alpha'\right)}{\sin\left(\left(\frac{(2w+1)\pi}{2V_d} - \alpha'\right)/2\right)} \\
&= (-1)^w\left(\frac{\cos(V_d\alpha')}{2\sin^2\left(\frac{(2w+1)\pi}{4V_d} - \alpha'/2\right)} + \frac{V_d\sin\left(\frac{(2w+1)\pi}{4V_d} - (V_d+1/2)\alpha'\right)}{\sin\left(\frac{(2w+1)\pi}{4V_d} - \alpha'/2\right)}\right).
\end{aligned}
\tag{56}
$$

Substituting Eq. (56) into Eq. (55) gives Eq. (26).

The posterior variance can be computed as

$$
\begin{aligned}
\widetilde{s}^{(d)}_{[\boldsymbol{X},\boldsymbol{\sigma}]}(\boldsymbol{x}', \boldsymbol{x}') &= \widetilde{k}'' - \widetilde{\boldsymbol{k}}'^{\top}\left(\boldsymbol{K} + \sigma^2 \boldsymbol{I}_{2V_d}\right)^{-1} \widetilde{\boldsymbol{k}}' \\
&= \widetilde{k}'' - \widetilde{\boldsymbol{k}}'^{\top}\frac{1}{\sigma_0^2}a\left(\boldsymbol{I}_{2V_d} + b\boldsymbol{1}_{2V_d}\boldsymbol{1}_{2V_d}^{\top} + c\boldsymbol{c}\boldsymbol{c}^{\top}\right)\widetilde{\boldsymbol{k}}' \\
&= \widetilde{k}'' - \frac{1}{\sigma_0^2}a\left(\|\widetilde{\boldsymbol{k}}'\|^2 + b(\widetilde{\boldsymbol{k}}'^{\top}\boldsymbol{1}_{2V_d})^2 + c(\widetilde{\boldsymbol{k}}'^{\top}\boldsymbol{c})^2\right) \\
&= \frac{\sigma_0^2 V_d(V_d+1)(2V_d+1)}{3(\gamma^2+2V_d)} \\
&\quad - \frac{1}{\sigma_0^2}a\left\{\sigma_0^4\frac{4V_d^2}{(\gamma^2+2V_d)^2}\left(\frac{(V_d+1)(2V_d+1)}{6} + V_d\cos(2V_d\alpha')\right) + c\sigma_0^4\left(\frac{4V_d^2\cos V_d\alpha'}{\gamma^2+2V_d}\right)^2\right\} \\
&= \frac{\sigma_0^2 V_d(V_d+1)(2V_d+1)}{3(\gamma^2+2V_d)} - \sigma_0^2 a\frac{4V_d^2}{(\gamma^2+2V_d)^2}\frac{(V_d+1)(2V_d+1)}{6} \\
&\quad - \sigma_0^2 a\left\{\frac{4V_d^3\cos(2V_d\alpha')}{(\gamma^2+2V_d)^2} + c\frac{16V_d^4\cos^2 V_d\alpha'}{(\gamma^2+2V_d)^2}\right\} \\
&= \frac{\sigma_0^2 V_d(V_d+1)(2V_d+1)}{3(\gamma^2+2V_d)} - \sigma_0^2\frac{\gamma^2+2V_d}{(\gamma^2+2V_d)\sigma^2/\sigma_0^2+2V_d}\frac{4V_d^2}{(\gamma^2+2V_d)^2}\frac{(V_d+1)(2V_d+1)}{6} \\
&\quad - \sigma_0^2\frac{\gamma^2+2V_d}{(\gamma^2+2V_d)\sigma^2/\sigma_0^2+2V_d}\left\{\frac{4V_d^3\cos(2V_d\alpha')}{(\gamma^2+2V_d)^2} - \frac{1}{(\gamma^2+2V_d)\sigma^2/\sigma_0^2+4V_d}\frac{8V_d^4(1+\cos 2V_d\alpha')}{(\gamma^2+2V_d)^2}\right\}
\end{aligned}
$$

$$= \sigma^2 \frac{V_d(V_d+1)(2V_d+1)}{3((\gamma^2+2V_d)\sigma^2/\sigma_0^2+2V_d)} - \sigma^2 \frac{4V_d^3\cos\left(2V_d\alpha'\right)}{((\gamma^2+2V_d)\sigma^2/\sigma_0^2+2V_d)((\gamma^2+2V_d)\sigma^2/\sigma_0^2+4V_d)}$$

$$- \sigma_0^2 \frac{8V_d^4(\cos\left(2V_d\alpha'\right)-1)}{(\gamma^2+2V_d)((\gamma^2+2V_d)\sigma^2/\sigma_0^2+2V_d)((\gamma^2+2V_d)\sigma^2/\sigma_0^2+4V_d)}, \tag{57}$$

which gives Eq. (27). □

## C.4 Proof of Theorem 3.2

In the first order case with $V_d = 1, \forall d = 1, \ldots, D$, the test VQE kernels for predicting derivatives $\partial_d f(\boldsymbol{x}'), \partial_d f(\boldsymbol{x}'')$ are

$$\widetilde{k}(\boldsymbol{x}, \boldsymbol{x}') = \frac{\partial}{\partial x_d'} k(\boldsymbol{x}, \boldsymbol{x}') = \sigma_0^2 \left( \frac{2\sin\left(x_d - x_d'\right)}{\gamma^2+2} \right) \prod_{d'\neq d} \left( \frac{\gamma^2 + 2\cos\left(x_{d'} - x_{d'}'\right)}{\gamma^2+2} \right),$$

$$\widetilde{k}(\boldsymbol{x}', \boldsymbol{x}'') = \frac{\partial^2}{\partial x_d' \partial x_d''} k(\boldsymbol{x}', \boldsymbol{x}'') = \sigma_0^2 \left( \frac{2\cos\left(x_d' - x_d''\right)}{\gamma^2+2} \right) \prod_{d'\neq d} \left( \frac{\gamma^2 + 2\cos\left(x_{d'}' - x_{d'}''\right)}{\gamma^2+2} \right).$$

Then, the kernels with the two training points $\boldsymbol{X} = (\boldsymbol{x}' - \alpha\boldsymbol{e}_d, \boldsymbol{x}' + \alpha\boldsymbol{e}_d)$ and the one test point $\boldsymbol{x}'$ are

$$\boldsymbol{K} = \sigma_0^2 \begin{pmatrix} 1 & \frac{\gamma^2+2\cos 2\alpha}{\gamma^2+2} \\ \frac{\gamma^2+2\cos 2\alpha}{\gamma^2+2} & 1 \end{pmatrix}, \qquad \widetilde{\boldsymbol{k}}' = \frac{2\sigma_0^2 \sin\alpha}{\gamma^2+2} \begin{pmatrix} -1 \\ 1 \end{pmatrix}, \qquad \widetilde{k}'' = \frac{2\sigma_0^2}{\gamma^2+2}.$$

With these kernels, the posterior mean is

$$\widetilde{\mu}^{(d)}_{[\boldsymbol{X},\boldsymbol{y},\boldsymbol{\sigma}]}(\boldsymbol{x}') = \widetilde{\boldsymbol{k}}'^{\top} \left( \boldsymbol{K} + \sigma^2\boldsymbol{I}_N \right)^{-1} \boldsymbol{y}$$

$$= \frac{2\sin\alpha}{\gamma^2+2} \begin{pmatrix} -1 & 1 \end{pmatrix} \begin{pmatrix} 1+\sigma^2/\sigma_0^2 & \frac{\gamma^2+2\cos 2\alpha}{\gamma^2+2} \\ \frac{\gamma^2+2\cos 2\alpha}{\gamma^2+2} & 1+\sigma^2/\sigma_0^2 \end{pmatrix}^{-1} \boldsymbol{y}$$

$$= \frac{2\sin\alpha}{\gamma^2+2} \begin{pmatrix} -1 & 1 \end{pmatrix} \frac{1}{(1+\sigma^2/\sigma_0^2)^2 - \left(\frac{\gamma^2+2\cos 2\alpha}{\gamma^2+2}\right)^2} \begin{pmatrix} 1+\sigma^2/\sigma_0^2 & -\frac{\gamma^2+2\cos 2\alpha}{\gamma^2+2} \\ -\frac{\gamma^2+2\cos 2\alpha}{\gamma^2+2} & 1+\sigma^2/\sigma_0^2 \end{pmatrix} \boldsymbol{y}$$

$$= \frac{2\sin\alpha}{\gamma^2+2} \frac{1}{(1+\sigma^2/\sigma_0^2) - \left(\frac{\gamma^2+2\cos 2\alpha}{\gamma^2+2}\right)} \begin{pmatrix} -1 & 1 \end{pmatrix} \boldsymbol{y}$$

$$= 2\sin\alpha \frac{y_2 - y_1}{(1+\sigma^2/\sigma_0^2)(\gamma^2+2) - (\gamma^2+2\cos 2\alpha)}$$

$$= \frac{(y_2 - y_1)\sin\alpha}{(\gamma^2/2+1)\sigma^2/\sigma_0^2 + 2\sin^2\alpha}.$$

The posterior variance is

$$\widetilde{s}^{(d)}_{[\boldsymbol{X},\boldsymbol{\sigma}]}(\boldsymbol{x}', \boldsymbol{x}') = \widetilde{k}'' - \widetilde{\boldsymbol{k}}'^{\top} \left( \boldsymbol{K} + \sigma^2\boldsymbol{I}_N \right)^{-1} \widetilde{\boldsymbol{k}}'$$

$$= \frac{2\sigma_0^2}{\gamma^2+2} - \frac{4\sigma_0^2 \sin^2\alpha}{(\gamma^2+2)^2} \begin{pmatrix} -1 & 1 \end{pmatrix} \begin{pmatrix} 1+\sigma^2/\sigma_0^2 & \frac{\gamma^2+2\cos 2\alpha}{\gamma^2+2} \\ \frac{\gamma^2+2\cos 2\alpha}{\gamma^2+2} & 1+\sigma^2/\sigma_0^2 \end{pmatrix}^{-1} \begin{pmatrix} -1 \\ 1 \end{pmatrix}$$

$$= \frac{2\sigma_0^2}{\gamma^2+2} - \frac{4\sigma_0^2 \sin^2\alpha}{(\gamma^2+2)^2} \frac{1}{(1+\sigma^2/\sigma_0^2) - \left(\frac{\gamma^2+2\cos 2\alpha}{\gamma^2+2}\right)} \begin{pmatrix} -1 & 1 \end{pmatrix} \begin{pmatrix} -1 \\ 1 \end{pmatrix}$$

$$= \frac{2\sigma_0^2}{\gamma^2+2} \left( 1 - \frac{4\sin^2\alpha}{(\gamma^2+2)\sigma^2/\sigma_0^2 + 2 - 2\cos 2\alpha} \right)$$

$$= \frac{2\sigma_0^2}{\gamma^2+2} \left( \frac{(\gamma^2+2)\sigma^2/\sigma_0^2}{(\gamma^2+2)\sigma^2/\sigma_0^2 + 4\sin^2\alpha} \right)$$

$$= \frac{\sigma^2}{(\gamma^2/2+1)\sigma^2/\sigma_0^2 + 2\sin^2\alpha}.$$

Thus we obtained Eq.(16). □

**Algorithm 1: (SGD-GradCoRe)** Improved SGD algorithm using a VQE-derivative kernel GP with the GradCoRe measurement selection subroutine, as described in Algorithm 2. The algorithm finds the minimum number of shots required to estimate the gradient wrt. parameter configurations $\hat{\boldsymbol{x}}$ of the quantum circuit to optimize with SGD. The optimization stops when the total number of measurement shots reaches the maximum number of observation shots allowed, i.e., $N_{\text{tot}-\text{shots}}$. To avoid cluttering notation, the algorithm is restricted to the case where $V_d = 1$. Generalization to an arbitrary $V_d$ is straightforward.

---

**Input** :
  - $\hat{\boldsymbol{x}}^0$: initial starting point (best point)

**Parameters** :
  - $V_d = 1$
  - $D$ : number of parameters to optimize, i.e., $\hat{\boldsymbol{x}} \in \mathbb{R}^D$.
  - $N_{\text{tot-shots}}$ : Total # of shots, i.e., maximum allowed quantum computing budget.
  - $C^{*2}$ : measurement variance using a single shot.
  - $\kappa_0$ : Initial GradCoRe threshold at step $t = 0$.
  - $T_{\text{initial}}$ : Number of steps in beginning to use initial GradCoRe threshold $\kappa_0$.
  - $c_0$ : smallest allowed GradCoRe threshold
  - $c_1$ : GradCoRe threshold scaling parameter

**Output** :
  - $\hat{\boldsymbol{x}}^*$ : optimal choice of parameters for the quantum circuit.

---

1 $n \leftarrow 0$ /* initialize consumed shot budget                              */
2 $t \leftarrow 0$ /* initialize optimization step                               */

3 $\boldsymbol{\kappa}^0 \leftarrow \mathbf{1}_D \kappa_0$ /* initial $\kappa_0$ to use for $T_{\text{initial}}$ steps          */
4 $\boldsymbol{X}^0, \boldsymbol{y}^0, \boldsymbol{\sigma}^0 \leftarrow (), (), ()$ /* initialize empty Gaussian process       */

5 **while** $n < N_{\text{tot-shots}}$ **do**
6  $\quad$ /* choose measurement points & number of shots s.t. $\hat{\boldsymbol{x}}^t$ is in the
  $\quad\quad$ GradCoRe of $\boldsymbol{\kappa}^t$                                           */
7  $\quad \check{\mathbf{X}}, \widetilde{\boldsymbol{\nu}} \leftarrow \texttt{gradcore\_measurements}(\boldsymbol{X}^t, \boldsymbol{y}^t, \boldsymbol{\sigma}^t, \hat{\boldsymbol{x}}^t, \boldsymbol{\kappa}^t)$ /* (Algorithm 2)       */

8  $\quad$ **for** $i \in \{1, ..., |\check{\mathbf{X}}|\}$ **do**
9  $\quad\quad \check{y}_i \leftarrow \texttt{quantum\_circuit}(\texttt{parameters=}\check{\mathbf{X}}_i\texttt{, shots=}\widetilde{\boldsymbol{\nu}}_i)$ /* measure chosen
  $\quad\quad\quad$ points                                                         */
10 $\quad\quad \check{\sigma}_i \leftarrow \frac{\overline{\sigma}^{*2}}{\widetilde{\boldsymbol{\nu}}_i}$
11 $\quad$ **end**
12 $\quad \check{\boldsymbol{y}}, \check{\boldsymbol{\sigma}} \leftarrow (\check{y}_1, ..., \check{y}_{|\check{\mathbf{X}}|}), (\check{\sigma}_1, ..., \check{\sigma}_{|\check{\mathbf{X}}|})$ /* concatenate observed values & noise   */
13 $\quad \boldsymbol{X}^{t+1}, \boldsymbol{y}^{t+1}, \boldsymbol{\sigma}^{t+1} \leftarrow (\boldsymbol{X}^t, \check{\boldsymbol{X}}), (\boldsymbol{y}^t, \check{\boldsymbol{y}}), (\boldsymbol{\sigma}^t, \check{\boldsymbol{\sigma}})$ /* add new observations to
  $\quad\quad$ Gaussian process                                                    */

14 $\quad \hat{\boldsymbol{x}}^{t+1} \leftarrow \hat{\boldsymbol{x}}^t - \rho \widetilde{\mu}_{[\boldsymbol{X}^{t+1}, \boldsymbol{\sigma}^{t+1}, \boldsymbol{y}^{t+1}]}(\hat{\boldsymbol{x}}^t)$ /* SGD (or variant) step using GP
  $\quad\quad$ derivative                                                          */
15 $\quad$ **if** $t \geq T_{\text{intial}}$ **then**
16 $\quad\quad \boldsymbol{\kappa}^{t+1} \leftarrow \mathbf{1}_D \max\left[c_0, \frac{c_1}{D} \sum_{d=1}^D \left(\widetilde{\mu}^{(d)}_{[\boldsymbol{X}^{t+1}, \boldsymbol{y}^{t+1}, \boldsymbol{\sigma}^{t+1}]}(\widehat{\boldsymbol{x}^t})\right)^2\right]$ /* adapt GradCoRe
  $\quad\quad\quad$ threshold                                                       */
17 $\quad$ **end**
18 $\quad t \leftarrow t + 1$ /* update the step                                    */
19 $\quad n \leftarrow n + \sum_d \widetilde{\boldsymbol{\nu}}_d$ /* update the consumed shot budget             */
20 **end**
21 **return** $\hat{\boldsymbol{x}}^*$

---

# D  ALGORITHM DETAILS

## D.1  GRADCORE PSEUDO-CODE

Algorithm 1 describes SGD-GradCoRe in detail. SGD-GradCoRe uses the GradCoRe measurement selection subroutine described in Algorithm 2, which selects measurement points and respective minimum required number of shots to estimate the quantum circuit parameter derivative required for

---

**Algorithm 2: (GradCoRe measurement selection subroutine)** Select the points to measure and respective minimum number of required shots such that when updating the GP with these new measurements, the GP's derivative uncertainty at the current best point is smaller than the threshold $\kappa$, i.e., the current point is within the GradCoRe.

---

**Input** :
- $\boldsymbol{X}, \boldsymbol{y}, \boldsymbol{\sigma}$ : Gaussian process at current step
- $\hat{\boldsymbol{x}}$ : current best point
- $\boldsymbol{\kappa} = (\kappa_1^2, \ldots, \kappa_D^2)$ : GradCoRe thresholds at current step

**Parameters** :
- $V_d = 1$
- $\overline{\sigma}^{*2}$ : measurement variance using a single shot.
- $\hat{\alpha}$ : shift from best point at the previous step (default to $\hat{\alpha} = \frac{\pi}{2}$)

**Output** :
- $\breve{\boldsymbol{X}}$ : points which should be measured and added to the GP to compute the derivative.
- $\widetilde{\boldsymbol{\nu}}$ : number of shots for the measured points.

1 **begin**
2    **for** $d \in \{1, ..., D\}$ **do**
3       $\breve{\boldsymbol{X}}_d \leftarrow (\hat{\boldsymbol{x}} - \hat{\alpha} \cdot \boldsymbol{e}_d, \hat{\boldsymbol{x}} + \hat{\alpha} \cdot \boldsymbol{e}_d)$ /* choose points to measure along d       */
4       $\breve{\sigma}_\pm \leftarrow \kappa_d$ /* initialize measurement noise to minimum (most
          expensive, $\kappa_d \ll \overline{\sigma}^*$)           */
5       **for** $\tilde{\sigma} \in [\sqrt{2}\kappa_d, \overline{\sigma}^*]$ **do**
6          /* create temporary GP copies, add points with $\tilde{\sigma}$ observation
            noise          */
7          $\boldsymbol{X}', \boldsymbol{y}', \boldsymbol{\sigma}' \leftarrow (\boldsymbol{X}, \breve{\boldsymbol{X}}_d), (\boldsymbol{y}, 0, 0), (\boldsymbol{\sigma}, \tilde{\sigma}, \tilde{\sigma})$
8          /* find largest observation noise for which $\hat{\boldsymbol{x}}$ is in the
            GradCoRe          */
9          **if** $(\widetilde{s}_{[\boldsymbol{X}', \boldsymbol{\sigma}']}(\hat{\boldsymbol{x}}, \hat{\boldsymbol{x}}) \leq \kappa_d^2) \wedge (\breve{\sigma}_\pm > \tilde{\sigma})$ **then**
10             $\breve{\sigma}_\pm \leftarrow \tilde{\sigma}$
11          **end**
12       **end**
13       $\widetilde{\boldsymbol{\nu}}_d \leftarrow \left(\frac{\overline{\sigma}^{*2}}{\breve{\sigma}_\pm}, \frac{\overline{\sigma}^{*2}}{\breve{\sigma}_\pm}\right)$ /* compute shots from variance through single shot
         variance $\overline{\sigma}^{*2}$          */
14    **end**
15    $\breve{\boldsymbol{X}} \leftarrow \left(\breve{\boldsymbol{X}}_1, \ldots, \breve{\boldsymbol{X}}_D\right)$ /* concatenate points to measure       */
16    $\widetilde{\boldsymbol{\nu}} \leftarrow (\widetilde{\boldsymbol{\nu}}_1, \ldots, \widetilde{\boldsymbol{\nu}}_D)$ /* concatenate shots to measure per point       */
17    **return** $\breve{\boldsymbol{X}}, \widetilde{\boldsymbol{\nu}}_d^{t+1}$
18 **end**

---

the SGD. Note that the grid search range in Line 5 in Algorithm 2 is set based on Corollary C.4 in Appendix C.

### D.2 PARAMETER SETTING

Every algorithm used in our benchmarking analysis has several hyperparameters to be set. For transparency and to allow the reproduction of our experiments, we detail the choice of parameters for EMICoRe, SubsCoRe and GradCoRe in Table 1. The SGLBO results were obtained using the original code from Tamiya and Yamasaki (2022) and we used the default setting from the original paper. For NFT, Bayes-NFT and Bayes-SGD runs, we used the default parameters specified in Table 2. For algorithmic efficiency, all Bayesian-SMO methods use the inducer option introduced in Nicoli et al. (2023a), retaining only the last $R \cdot 2V_d \cdot D - 1 = 399$ measured points once more than $R \cdot 2V_d \cdot D - 1 + D = 439$ points were stored in the GP, where we chose $R = 5$. Since the discarded points are replaced with a single point predicted from them, the number of the traininig points for the GP is kept constant at $R \cdot 2V_d \cdot D = 400$. On the other hand, Bayesian-SGD methods measure (at most, in the SGD-GradCoRe case) $2V_d D = 80$ points per SGD step, and we retain $R \cdot 2V_d \cdot D = 400$ points after more than $(R + 1) \cdot 2V_d \cdot D = 480$ points are measured. Unlike the Bayesian-SMO

Table 1: Algorithm specific parameter choice for EMICoRe, SubsCoRe and GradCoRe for all experiments (unless specified otherwise).

| | **Algorithmic specific parameters** | |
|---|---|---|
| `--acq-params` | **EMICoRe params** | as in Nicoli et al. (2023a) |
| `func` | `ei` | Base acq. func. type |
| `optim` | `emicore` | Optimizer type |
| `pairsize` $(J_{SG})$ | `20` | # of candidate points |
| `gridsize` $(J_{OG})$ | `100` | # of evaluation points |
| `corethresh-strategy` | `grad` | Gradient strategy for $\kappa$ |
| `pnorm` | `2` | Order of gradient norm |
| `corethresh` $(\kappa)$ | `256` | CoRe threshold $\kappa$ |
| `corethresh_width` $(T_{initial})$ | `40` | # initial steps with fixed $\kappa$ |
| `coremin_scale` $(C_0)$ | `2048` | Coefficient $C_0$ for updating $\kappa$ |
| `corethresh_scale` $(C_1)$ | `1.0` | Coefficient $C_1$ for updating $\kappa$ |
| `stabilize_interval` | `41` | Stabilization interval in SMO steps |
| `samplesize` $(N_{MC})$ | `100` | # of MC samples |
| `smo-steps` $(T_{NFT})$ | `0` | # of initial NFT steps |
| `smo-axis` | `True` | Sequential direction choice |
| `--acq-params` | **SubsCoRe params** | as in Anders et al. (2024) |
| `optim` | `subscore`[5] | Optimizer type |
| `readout-strategy` | `center` | Alg type SubsCoRe |
| `corethresh-strategy` | `grad` | Gradient strategy for $\kappa$ |
| `pnorm` | `2` | Order of gradient norm |
| `corethresh` $(\kappa)$ | `256` | Initial $N_{shots}$ for CoRe |
| `corethresh_width` $(T_{initial})$ | `40` | # initial steps with fixed $\kappa$ |
| `coremin_scale` $(C_0)$ | `2048` | Coefficient $C_0$ for updating $\kappa$ |
| `corethresh_scale` $(C_1)$ | `1.0` | Coefficient $C_1$ for updating $\kappa$ |
| `stabilize_interval` | `41` | Stabilization interval in SMO steps |
| `coremetric` | `readout` | Metric to set CoRe |
| `--acq-params` | **GradCoRe params** | this paper[6] |
| `optim` | `gradcore` | Optimizer type |
| `corethresh-strategy` | `grad` | Gradient strategy for $\kappa$ |
| `pnorm` | `2` | Order of gradient norm |
| `corethresh` $(\kappa)$ | `256` | Initial $N_{shots}$ for CoRe |
| `corethresh_width` $(T_{initial})$ | `40` | # initial steps with fixed $\kappa$ |
| `coremin_scale` $(C_0)$ | `2048` | Coefficient $C_0$ for updating $\kappa$ |
| `corethresh_scale` $(C_1)$ | `1.4` | Coefficient $C_1$ for updating $\kappa$ |
| `coremetric` | `readout` | Metric to set CoRe |
| `lr` | `0.05` | learning rate for SGD |
| `gdoptim` | `adam` | Optimizer for SGD |

methods, we do not add additional inducer based on the prediction from the discarded points, and therefore the number of the training points for the GP is kept constant at $R \cdot 2V_d \cdot D = 400$.

# E  EXPERIMENTAL DETAILS

As discussed in the main text, our experiments focus on the same experimental setup as in Nicoli et al. (2023a) and Anders et al. (2024). Specifically, starting from the quantum Heisenberg Hamiltonian, we reduce it to the special case of the Ising Hamiltonian at the critical point by choosing the suitable couplings, namely

Ising Hamiltonian at criticality: $J = (-1.0, 0.0, 0.0)$; $h = (0.0, 0.0, -1.0)$.

---

[5]a.k.a., "*readout*" in Nicoli et al. (2023a).

[6]All hyperparameters not specified in the table are set to the default in Nicoli et al. (2023a).

Table 2: Default choice of circuit parameters and kernel hyperparameters for all experiments (unless specified otherwise).

|  | **Deafult params** |  |
|---|---|---|
| --n-qbits | 5 | # of qubits |
| --n-layers | 3 | # of circuit layers |
| --circuit | esu2 | Circuit name |
| --pbc | False | Open Boundary Conditions |
| --n-iter | 1*10**7 | # max number of readouts |
| --kernel | vqe | Name of the kernel |

| --kernel-params | **Bayes-NFT** | **EmiCoRe** | **SubsCoRe** | **GradCore** | **Bayes-SGD** |
|---|---|---|---|---|---|
| gamma | 3 | 3 | 3 | 3 | 3 |
| sigma_0 | 10 | 10 | 10 | 10 | 10 |

It is important to note that due to the finite size of the system at hand, this choice of parameters does not imply criticality but already represents a challenging setup, as discussed in Sec. I.2 in Nicoli et al. (2023a). We stop the optimization when a predetermined maximum number of cumulative shots (total measurement budget on the quantum computer) is reached; unless specified otherwise, we set this cutoff to $N_{\text{shots}}^{\max} = 1 \cdot 10^7$.

Our implementation of GradCoRe can be found in the supplementary zip file and will be made available on Github upon acceptance. In our experiments, the kernel parameters $\sigma_0$ and $\gamma$ are fixed to the values in Table 2. Furthermore, NFT, Bayes-NFT, Bayes-SGD, SubsCoRe and GradCoRe require fixed shifts for the points to measure at each iteration. In our experiments, we always used $\alpha = \frac{2\pi}{3}$ for SMO based methods (as this makes the uncertainty uniform in the 1D-subspace, as discussed in Anders et al. (2024)), and $\alpha = \frac{\pi}{2}$ for SGD based methods (as this minimizes the uncertainty in the noisy case, as discussed in Section 3), unless explicitly stated otherwise.

Each experiment shown in the paper was repeated 100 times (trials) with differently seeded starting points. We aggregated the statistics from these independent trials and presented them in our plots. We used the same starting point for every algorithm in each trial to ensure a fair comparison between all approaches. Note that SGD-based methods do not require measurements at the starting point, but SMO-based methods do. Therefore, each starting point is further paired with a fixed initial measurement.

All experiments were conducted on Intel Xeon Silver 4316 @ 2.30GHz CPUs.

# F  DETAILED BEHAVIOR OF BAYESIAN PSR AND GRADCORE

Figure 7 shows the gradient esitmation error by PSR (dashed curve) and Bayesian PSR (solid curve) during the SGD optimization. We clearly observe that Bayesian PSR estimates the gradient more accurately than PSR.

Figure 8 shows the behavior of the GradCoRe threshold $\kappa(t)$ (left), and the number $\nu(t)$ of measurement shots (left) that GradCoRe used in each SGD iteration.

## F.1  ADDITIONAL RESULTS

Figures 9–11 show the optimization result for the Ising Hamiltonian with $(Q = 7)$-qubits $(L = 5)$-layers circuit, that for the Heisenberg Hamiltonian with $(Q = 5)$-qubits $(L = 3)$-layers circuit, and that for the Heisenberg Hamiltonian with $(Q = 7)$-qubits $(L = 5)$-layers circuit. We observe that the advantages of GradCoRe are more prominent for larger qubits.

To confirm the statistical significance that the GradCoRe outperforms the baselines, we applied the Wilcoxon signed rank test. Table 3 shows the p-values for the null hypothesis that there is no

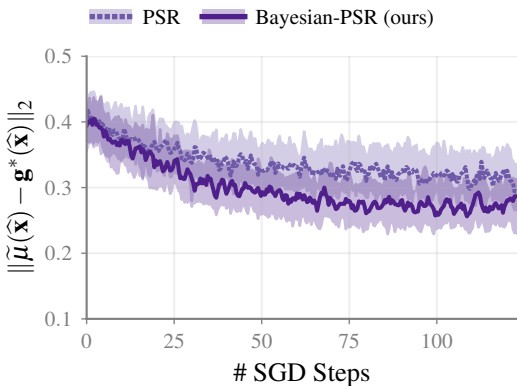

Figure 7: Gradient estimation error by PSR (dashed curve) and Bayesian PSR (solid curve) for $N_{\text{shots}} = 1024$, evaluated by the L2-distance between the estimated gradient $\widetilde{\boldsymbol{\mu}}(\widehat{\boldsymbol{x}})$ and the true gradient $\boldsymbol{g}^*(\widehat{\boldsymbol{x}})$ (computed by the PSR with simulated noiseless measurements).

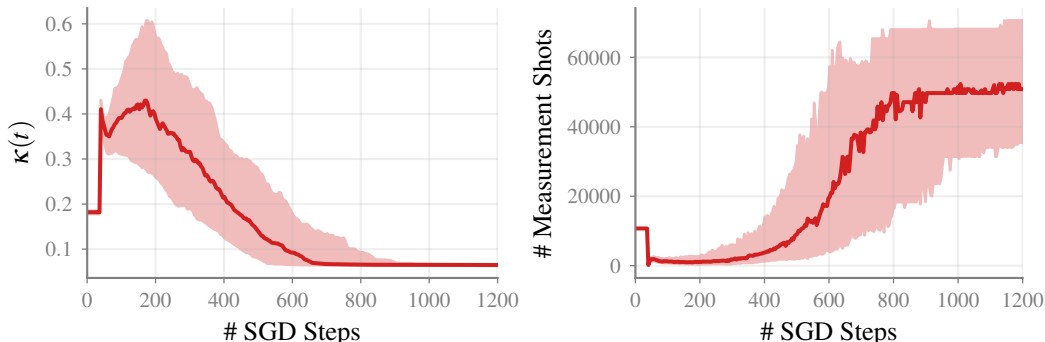

Figure 8: The GradCoRe threshold $\kappa(t)$ (left), set according to Eq. (19), and the number of measurement shots (right) per SGD iteration used by GradCoRe. As expected, the number of shots gradually increases as the GradCoRe threshold decreases, reflecting the flatness of the objective function via the gradient norm estimation.

difference in performance between GradCoRe and the baselines. In all cases, we observe p-values smaller than 0.05, proving the advantage of GradCoRe.

## G    PERFORMANCE EVALUATION OF GRADIENT INFORMATION WITH BO (GIBO)

As mentioned in Section 1, the gradient information with BO (GIBO) (Müller et al., 2021) can be considered as a general BO baseline. Here we compare its performance with other baseline methods. For GIBO, we use the VQE kernel (Nicoli et al., 2023a) but *do not* use the strong prior information about the optimal locations of observed points given by Theorem 3.2. Instead, we minimize the trace of the predictive variance, as an acquisition function, following Algorithm 1 in Müller et al. (2021). Specifically, we sequentially optimize $2D$ points by using the L-BFGS-B (Fletcher, 2000) optimizer in each SGD iteration. For choosing each point, we start from $10 \cdot D$ initial points drawn from the uniform distribution on the $D$-dimensional sphere with radius $\delta = 0.2$ centered at the current best point, and choose the best one (see Table 1 in Appendix A.8 in Müller et al. (2021)). Figure 12 compares the performance of GIBO with the other baseline methods evaluated in the main text, i.e., SGD-PSR and NFT with $N_{\text{shot}} = 1024$, as well as our proposed GradCoRe. We see that GIBO is outperformed by both the the plain SGD-PSR and NFT, even though it uses some physical prior information through the VQE kernel.

The typical solvers for VQEs, including the state-of-the-art methods, use the strong physical knowledge for determining the observed locations—they observe the points at $\widehat{\boldsymbol{x}} \pm \frac{\pi}{2}\boldsymbol{e}_d$ for $d = 1, \ldots, D$.

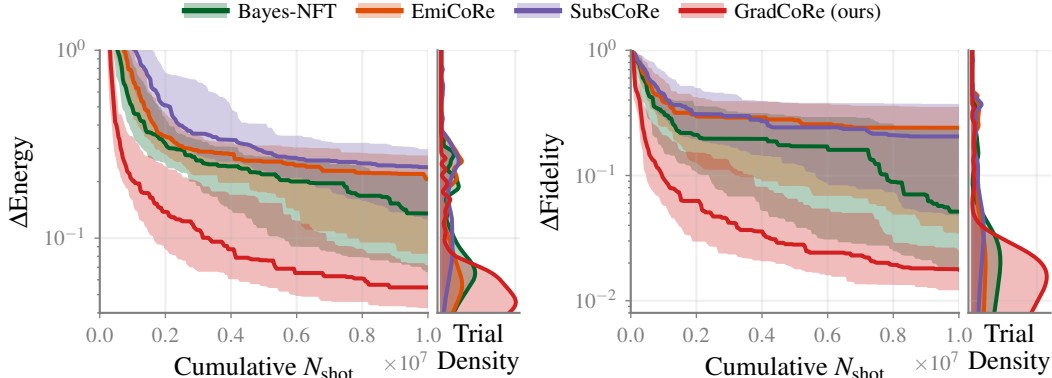

Figure 9: Energy (left) and Fidelity (right) achieved with the cumulative number of measurement shots for the Ising Hamiltonian with $(Q = 7)$-qubits $(L = 5)$-layers quantum circuit.

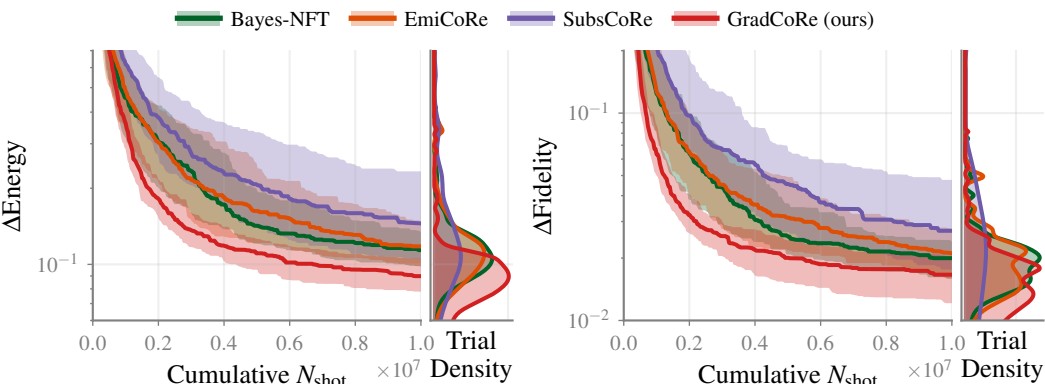

Figure 10: Energy (left) and Fidelity (right) achieved with the cumulative number of measurement shots for the Heisenberg Hamiltonian with $(Q = 5)$-qubits $(L = 3)$-layers quantum circuit.

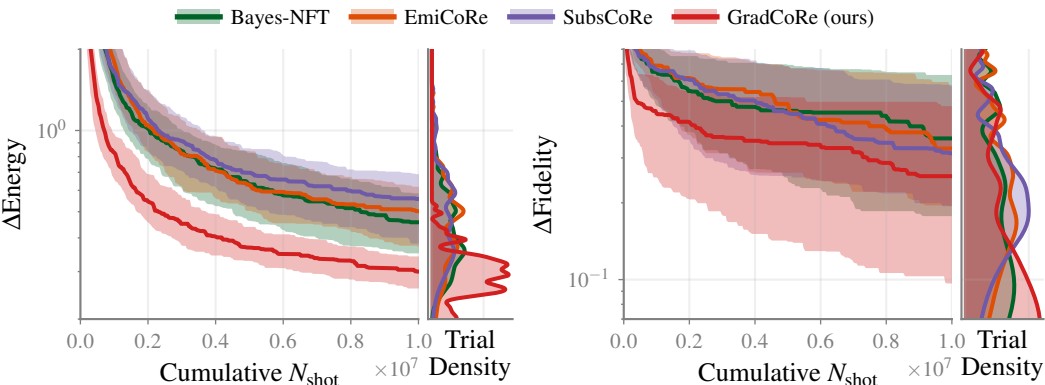

Figure 11: Energy (left) and Fidelity (right) achieved with the cumulative number of measurement shots for the Heisenberg Hamiltonian with $(Q = 7)$-qubits $(L = 5)$-layers quantum circuit.

In this paper, we proved in Theorem 3.2 that this choice is optimal for minimizing the uncertainty of gradient estimation. Since we use the VQE kernel also for GIBO, its poor performance is due to the inaccurate optimization of the acquisition function in $D$-dimensional space for $2D$ points, which is known to be challenging (Frazier, 2018). On the contrary, our GradCoRe observes the *theoretically optimal locations*, based on our Theorem 3.2, instead of tackling the challenging optimization of the multipoint acquisition function. This in principle may be achieved with GIBO equipped with an *oracle multi-points acquisition function optimizer, which does not exist*. This further demonstrates that

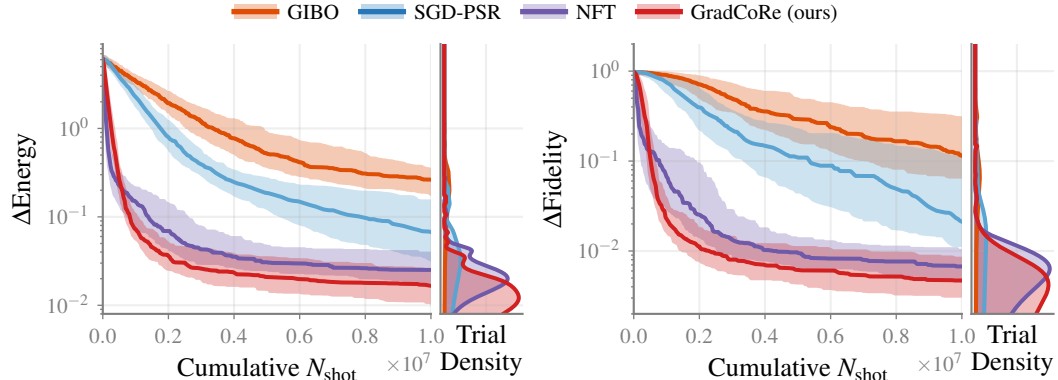

Figure 12: Performance comparison between GIBO (with VQE kernel), SGD-PSR, NFT, and GradCoRe.

combining existing techniques developed separately in the machine learning and physics literature can enhance performance.

Table 3: Obtained p-values by performing the Wilcoxon signed rank test. Each column indicates the baseline against which GradCoRe is compared, showing the p-values obtained in different settings. All p-values are smaller than 0.05, implying consistent performance improvement by GradCoRe over all settings.

| | $\Delta$**Energy** | | | $\Delta$**Fidelity** | | |
|---|---|---|---|---|---|---|
| **Experiment** | **Bayes-NFT** | **EmiCoRe** | **SubsCoRe** | **Bayes-NFT** | **EmiCoRe** | **SubsCoRe** |
| Ising (3,5) | 9.68e-05 | 4.23e-09 | 3.84e-05 | 1.80e-02 | 1.42e-07 | 5.19e-04 |
| Ising (5,7) | 1.16e-07 | 4.02e-10 | 7.49e-10 | 1.54e-05 | 4.06e-09 | 2.89e-08 |
| Heis. (3,5) | 8.57e-11 | 8.19e-11 | 2.29e-12 | 5.30e-06 | 2.78e-07 | 3.16e-10 |
| Heis. (5,7) | 2.37e-16 | 5.84e-16 | 2.82e-17 | 1.31e-02 | 2.69e-02 | 8.20e-03 |

## H CLASSICAL COMPUTATION COST

Following common practices as in previous work on VQE optimization, we focus only on the quantum computation cost when evaluating the optimization performance in Section 5. Here we discuss that the classical computation cost is still negligible even when the gradient is estimated by GP. The cost of GP prediction is dominated by the Cholesky inversion of the kernel matrix, which scales $\mathcal{O}(N^3)$ in the number of training points $N$. Therefore, the classical computation cost for estimating all elements of the gradient by Bayesian SGD and GradCoRe is $\mathcal{O}(D \cdot N^3)$, where the maximum number of training points is upper-bounded by $\overline{N} = R \cdot 2Vd \cdot D = 400$ (see Appendix D.2 for the choice of $N$ at each step).

Table 4 reports on the classical computation time per step in comparison with the SGD with the standard PSR. Although GP prediction slows down Bayes-SGD by two orders of magnitude, the computation time is still small, and thus it is reasonable to ignore it in the context of VQE optimization.

| Algorithm | SGD (with standard PSR) | Bayes-SGD |
|---|---|---|
| $\langle t_{step} \rangle \pm \sigma_{t_{step}}$ | $(1.4 \pm 0.2) \cdot 10^{-5}$ | $(4.3 \pm 0.7) \cdot 10^{-3}$ |

Table 4: Wall-clock classical computation time per step for SGD with the standard PSR and Bayes-SGD for the Ising Hamiltonian, simulated using the `Efficient SU(2)` ansatz with $Q = 5$ qubits and $L = 3$ layers. Average and standard deviation are computed over a full optimization run, using $10^7$ shots.

# I GAUSSIANITY OF OBSERVATION NOISE

Here we empirically demonstrate that the observation noise approximately follows a Gaussian distribution. Figure 13 shows the distribution of energy observations with $N_{\text{shot}} = 1, 8, 128, 1024$ at a randomly chosen fixed point in the 40-dimensional parameter space of `Efficient SU(2)` circuits with $Q = 5$ qubits and $L = 3$ layers. Note that the center of the distribution is shifted to the origin by subtracting the average of the energy observations. Since every single measurement (shot) is a sum of multiple terms, the observation noise nearly follows a Gaussian distribution, even for $N_{\text{shot}} = 1$.

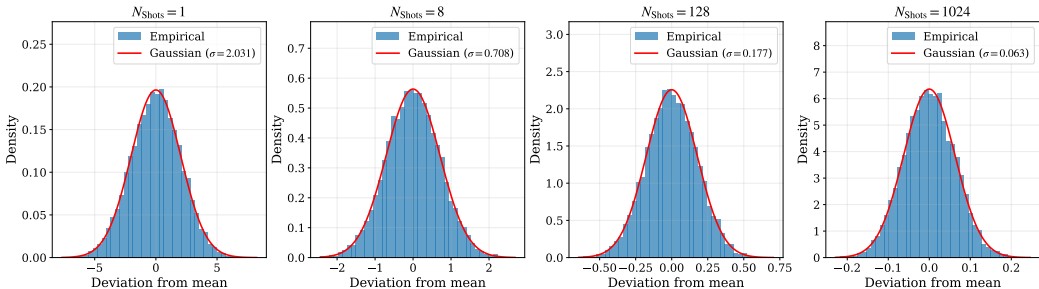

Figure 13: Observation noise distribution for $N_{\text{shot}} = 1, 8, 128, 1024$ at a randomly chosen fixed point.

