# OpenReview forum: "Bayesian Parameter Shift Rules in Variational Quantum Eigensolvers"
_ICLR.cc/2026/Conference — ICLR 2026 Poster_

### Official Review · Reviewer_fLQm · 2025-10-31

**Soundness:** 3
**Presentation:** 2
**Contribution:** 2
**Rating:** 4
**Confidence:** 2

**Summary:**

This work proposes the Bayesian parameter shift rule (PSR), which uses Gaussian process with VQE kernel, to estimate the gradient of objective in variational quantum eigensolver. Furthermore, guided by the posterior uncertainty, the stochastic gradient decent (SGD) step is completed in the gradient confident area. Numerical results demonstrate that the proposed optimization strategy can achieve faster convergence than the vanilla SGD.

**Strengths:**

- The manuscript provides a comprehensive theoretical analysis and numerical results, which are convincing.

**Weaknesses:**

- The proposed method is a natural application of Bayesian optimization with Gaussian process to VQE gradient estimation, which offers limited novelty.
- The motivation of employing the Bayesian PSR is not clearly clarified. It would be more convincing to discuss the challenges the proposed method tries to address.
- The manuscript lacks a formal statement or theorem guaranteeing the existence of an optimal sequence for both the shift parameter and shot allocation that ensures the fastest convergence.

**Questions:**

Please see the Weakness section

---

> ### Author Response · Authors · 2025-11-21
> **Author response to Reviewer fLQm**
>
> We thank the reviewer for their comments.  Please find our answers below.  We also uploaded a revised manuscript, where the revised/added sentences are highlighted in blue.
>
> >The proposed method is a natural application of Bayesian optimization with Gaussian process to VQE gradient estimation, which offers limited novelty.
>
> We respectfully disagree with the reviewer’s assessment regarding the limited novelty of our approach and would like to clarify the contributions more precisely. While our method indeed leverages Bayesian optimization and Gaussian processes for gradient estimation, this is only one component of the overall contribution.
>
> First, the analytical uncertainty expression derived in Theorem 3.2 enables us to establish the optimality of the shift $\alpha=\pi/2$, a result that, to the best of our knowledge, has not been shown before. This constitutes a novel theoretical insight into gradient estimation for VQE.
>
> Second, our proposed GradCoRe algorithm introduces a superior shot-adaptive strategy that consistently outperforms all baselines across multiple settings. This adaptive mechanism goes beyond a straightforward application of BO (see the comparison with GIBO in Appendix G), and is made possible by the uncertainty quantification and the optimal observation point configuration provided through our theoretical analysis.
>
> Taken together, these elements provide substantial theoretical and methodological novelty beyond merely applying Bayesian optimization with Gaussian processes to VQE gradient estimation.
>
> >The motivation of employing the Bayesian PSR is not clearly clarified. It would be more convincing to discuss the challenges the proposed method tries to address.
>
> We thank the reviewer for pointing this out and are happy to clarify the motivation behind employing Bayesian PSR. The baseline method—SGD combined with plain PSR—does not properly account for observation noise. As a consequence, gradient uncertainty is not incorporated in a principled way, and the number of measurement shots is typically fixed, which can be highly inefficient.
>
> Our aim is to address these limitations by introducing Bayesian PSR with Gaussian processes, which provides two key advantages: (i) the ability to reuse past observations, and (ii) access to uncertainty estimates for the gradient. While reusing observations does improve gradient estimation accuracy (as shown in Fig. 7 in Appendix F), it does not translate into substantial optimization improvements (Fig. 3).
>
> In contrast, the uncertainty quantification offered by Bayesian PSR enables adaptive control of the measurement cost, which directly tackles the inefficiency of using a fixed number of shots. This capability is essential for our proposed GradCoRe algorithm, which leverages uncertainty estimates to allocate shots more effectively and ultimately outperforms all baseline methods, including both SGD and SMO-based approaches. We highlight this motivation and its implications in Section 5.2.
>
> >The manuscript lacks a formal statement or theorem guaranteeing the existence of an optimal sequence for both the shift parameter and shot allocation that ensures the fastest convergence.
>
> We thank the reviewer for raising this point. Theoretical optimality analysis is an interesting research direction, but it is out of the scope of this paper.

---

### Official Review · Reviewer_9nAn · 2025-11-02

**Soundness:** 3
**Presentation:** 3
**Contribution:** 3
**Rating:** 8
**Confidence:** 2

**Summary:**

The authors propose a method for optimizing variational quantum eigensolvers. The method aims to reduce the number of measurement shots by estimating gradients using Gaussian Processes (GPs). This Bayesian approach allows the algorithm to reuse previous observations in its gradient estimates. The authors combine this with an adaptive observation cost strategy, "GradCoRe", and show that this combined method outperforms comparable approaches for a fixed measurement shot budget.

**Strengths:**

* The problem it targets—reducing measurement shots—is high-impact and central to near-term quantum algorithms.
* They obtain rigorous estimates for the mean and covariance of the gradient GP (Theorem 3.1) and show that their Bayesian PSR converges to the standard PSR in the noiseless limit.
* In a simplified setting (Theorem 3.2), they use this estimate to demonstrate that the common $\pi/2$ shift is optimal for minimizing uncertainty, justifying a technique used in the literature.
* The proposed method is a clever combination of existing tools (SGD, GPs, and physics-informed kernels).

**Weaknesses:**

* The method appears to be a combination of existing methods (which, as noted, is also a strength), making the novelty of the contribution somewhat unclear.
* The experiments seem fairly limited, primarily focusing on a $(Q=5)$-qubit system. A comparison on a higher-dimensional problem would be needed to substantiate the "state-of-the-art" claim.
* The graphs comparing techniques (e.g., Figure 4) are hard to parse. I would suggest experimenting with ways to represent the uncertainty without using shaded fills, which tend to obscure each other.
* The derivation of the derivative GP (Section 2.1) is essential to the method. While the paper states the kernel modifications and alludes to Appendix A, a clearer, self-contained (or better-referenced) derivation in the main text would be beneficial.
* While the paper focuses on quantum cost (measurement shots), it neglects the *classical* computation cost. The method requires GP regression and solving an optimization problem (Eq. 17) at each step. An empirical comparison of this classical overhead would be helpful.

**Questions:**

* Is it sensible or standard to assume that the observation noise in the model is Gaussian??
* The experimental setup (Ising model, $(Q=5)$, $(L=3)$) is taken from Nicoli et al. (2023a). Is this 5-qubit problem considered a sufficient benchmark for "state-of-the-art" claims in the field?

---

> ### Author Response · Authors · 2025-11-21
> **Author response to Reviewer 9nAn**
>
> We thank the reviewer for their positive assessment of our work. We answer their questions below. We also uploaded a revised version, addressing reviewers’ comments (the revised/added sentences are highlighted in blue).
>
> >Is it sensible or standard to assume that the observation noise in the model is Gaussian??
>
> As mentioned in Section 2.2 (see footnote 2), a single measurement of the Hamiltonian is a sum of multiple terms that are measured independently.  Furthermore, in many cases, a single observation is a sum of many measurement shots.  For this reason, the observation noise can typically be approximated with Gaussian noise, due to the central limit theorem. To validate this empirically, we plot the distribution for the energy measurement at a fixed random point for different numbers of measurement shots.    We observe that the distribution is already near Gaussian for a single shot measurement.
>
> >The experimental setup (Ising model, , ) is taken from Nicoli et al. (2023a). Is this 5-qubit problem considered a sufficient benchmark for "state-of-the-art" claims in the field?
>
> In Appendix F.1, Figs. 9-11 show results for larger system sizes (7-qubit, 5 layers) and also with the Heisenberg Hamiltonian.  There we observe that GradCoRe’s performance gains tend to become more pronounced as the problem size increases in terms of qubits or complexity of the Hamiltonian.  When it comes to state-of-the-art assessment, we compare GradCoRe with previous works benchmarking on these system sizes and prove GradCoRe evidently outperforms them. Therefore, we expect that GradCoRe’s advantage will persist for larger problems.
>
> >The method appears to be a combination of existing methods (which, as noted, is also a strength), making the novelty of the contribution somewhat unclear.
>
> Although our method integrates elements from existing approaches, this should not be taken to undermine the novelty of our contribution. Theorems 3.1 and 3.2 introduce the theoretical basis required for Bayesian PSR, which represent a substantial novelty in itself.
>
> >The graphs comparing techniques (e.g., Figure 4) are hard to parse. I would suggest experimenting with ways to represent the uncertainty without using shaded fills, which tend to obscure each other.
>
> We thank the reviewer for their feedback on our plots. To ensure a fair comparison with prior work, we adopted the same plotting style commonly used in the existing literature. We acknowledge, however, that the resulting visualizations may appear cluttered at times. Unfortunately, we were unable to identify a clearer alternative (e.g., avoiding shaded fills or different filling styles). We would welcome any suggestions the reviewer may have.
>
> >The derivation of the derivative GP (Section 2.1) is essential to the method. While the paper states the kernel modifications and alludes to Appendix A, a clearer, self-contained (or better-referenced) derivation in the main text would be beneficial.
>
>  In the revision, we updated the description that introduces the derivative kernels, Eqs. (4) and (5).  Specifically, we explain that we only need to adjust the kernel function, which corresponds to the covariance of GP prior, so that they are consistent with the original kernel function defined for non-derivative outputs.  We also refer to the analytic-forms of derivative kernels, Eqs.(38) and (39), in the first paragraph of Section 3.
>
> >While the paper focuses on quantum cost (measurement shots), it neglects the classical computation cost. The method requires GP regression and solving an optimization problem (Eq. 17) at each step. An empirical comparison of this classical overhead would be helpful.
>
> In the revision, we discuss the classical scaling and computation cost in Appendix H.  Bayes-SGD and GradCoRe need to perform GP predictions, with at most $N=400$ training points, D times in each iteration.  As shown in Table 4, the wall-clock time per optimization step is $\sim 1 \cdot 10^{-5}$ sec for SGD (with the standard PSR), and is $ \sim 4 \cdot 10^{-3}$ sec for Bayes-SGD and GradCoRe, i.e., the classical computation time of GradCoRe is 400 times larger than the SGD.  However, the classical computation time is still small and reasonable to be ignored in the context of VQE optimization.

---

### Official Review · Reviewer_TZCS · 2025-11-06

**Soundness:** 3
**Presentation:** 3
**Contribution:** 3
**Rating:** 6
**Confidence:** 3

**Summary:**

This paper presents a Bayesian variant of the well-known parameter shift rules (PSRs). Bayesian PSRs enable flexible gradient estimation from observations at arbitrary locations and unify the standard PSRs. Connecting the Bayesian PSRs with SGD, the paper proposes a new optimizer named gradient confident region (GradeCoRe), which outperforms many STOA methods for VQE training.

**Strengths:**

- Bayesian PSRs are a natural generalization of the standard PSRs, and they enable more efficient exploitation of previous measurement data by taking uncertainty information into account.
- The theoretical relationship between Bayesian PSR and existing PSRs is established. Bayesian PSR reduces to standard PSRs in certain cases. Moreover, the optimality of PSRs is established.
- Leveraging this new gradient Bayesian-based gradient estimation technique, gradient-based optimizers demonstrate improved numerical performance in various VQE problems.

**Weaknesses:**

- The current Bayesian PSR is derived based on Gaussian processes. Is this an appropriate assumption for VQE? How to justify the GP model?
- Is there a quantitative way to compare Bayesian PSRs and standard PSRs in terms of the quality of gradient estimation and sample complexity?

**Questions:**

- Is there an intuitive explanation for why the uncertainty dependence on $\alpha$ is a "U" shape curve (as in the last panel in Figure 2)?
- Even with Bayesian PSRs, it is a bit counterintuitive that an SGD-based algorithm can outperform Bayesian optimization methods. Is there an explanation for this?
- Is it possible to establish the optimality argument in the second order? If not, what are the challenges?
- Does the momentum information (e.g., momentum SGD + Bayesian PSRs) further accelerate the optimizer?

---

> ### Author Response · Authors · 2025-11-21
> **Author response to Reviewer TZCS**
>
> We thank the reviewer for their thorough review of our manuscript and for raising relevant questions. In the following we address the raised questions and concerns. We also uploaded a revised version, addressing reviewers’ comments (the revised/added sentences are highlighted in blue).
>
> >The current Bayesian PSR is derived based on Gaussian processes. Is this an appropriate assumption for VQE? How to justify the GP model?
>
> Under the assumption that the observation noise can be approximated by Gaussian (see the first answer to Reviewer 9nAn), we argue that GP with an appropriate kernel is a reasonable choice.  With the VQE kernel, GP is equivalent to the linear regression model with the trigonometric basis functions, which has the necessary and sufficient expressivity for general VQE problems. We refer to [Nicoli et. al (NeurIPS 2023)](https://proceedings.neurips.cc/paper_files/paper/2023/file/3adb85a348a18cdd74ce99fbbab20301-Paper-Conference.pdf) for further insights and theoretical proofs.
>
> >Is there a quantitative way to compare Bayesian PSRs and standard PSRs in terms of the quality of gradient estimation and sample complexity?
>
> In Fig. 7 in Appendix F, we directly compare the gradient estimation accuracy of the standard PSR and  Bayesian PSR, where the ground-truth gradient is obtained by Qiskit simulation in the infinite shot limit (exact measurement).  We observe that Bayesian PSR gives more accurate gradient estimation than the standard PSR. We emphasized this point in Section 5.2 in the revision.  We hope this answers the question, otherwise we would appreciate it if the reviewer could elaborate a bit more on what they mean by sample complexity in this context.
>
> >Is there an intuitive explanation for why the uncertainty dependence on 𝝰 is a "U" shape curve (as in the last panel in Figure 2)?
>
> The search domain is $[0, 2\pi)^D$ with periodic boundary conditions. Therefore, $\pm \pi/2$ are the largest span $(= \pi)$ observations.  One can intuitively understand that the maximum possible span gives the lowest uncertainty, similarly to the linear slope estimation.  We added this explanation in the caption of Fig. 2.
>
> >Even with Bayesian PSRs, it is a bit counterintuitive that an SGD-based algorithm can outperform Bayesian optimization methods. Is there an explanation for this?
>
> If the reviewer refers to EMICoRe as a Bayesian optimization (BO) method, we note that  [EMICoRe](https://proceedings.neurips.cc/paper_files/paper/2023/file/3adb85a348a18cdd74ce99fbbab20301-Paper-Conference.pdf) is an SMO-based method with fixed number of shots, and therefore, it is not very surprising that GradCoRe outperforms EMICoRe.  If the reviewer refers to [GIBO](https://proceedings.neurips.cc/paper/2021/file/ad0f7a25211abc3889cb0f420c85e671-Paper.pdf) in Appendix G, the weak performance of GIBO (even weaker than the plain NFT and SGD) is due to the difficulty of finding the optimal set of multiple (=$2 \cdot V_d \cdot D$) observation points.  There is a possibility that Bayesian optimization to fine-fune the observed locations of GradCoRe (which are chosen based on Theorem 3.2) could improve the performance, which we leave as future work.
>
> >Is it possible to establish the optimality argument in the second order? If not, what are the challenges?
>
> The optimality for estimating higher-order derivatives can be analyzed in a similar way by evaluating the uncertainty of the GP predictions for higher-order derivatives.  However, many more observed points lying in a higher-dimensional subspace are involved. Therefore, a theoretical analysis is much more complicated and out of the scope of this work.
>
> >Does the momentum information (e.g., momentum SGD + Bayesian PSRs) further accelerate the optimizer?
>
> As mentioned in the “Algorithm Setting” paragraph of Section 5, we use the ADAM optimizer, which already incorporates momentum information.

---

### Official Review · Reviewer_pQ8e · 2025-11-06

**Soundness:** 3
**Presentation:** 3
**Contribution:** 2
**Rating:** 6
**Confidence:** 3

**Summary:**

This paper introduces the Bayesian Parameter Shift Rule (Bayesian PSR), a method that uses Gaussian Processes (GPs) with a physics-informed VQE kernel to estimate the gradient of the VQE objective. A key feature of this Bayesian estimator is its flexibility, allowing it to reuse observations to improve the accuracy of the gradient estimate. The paper also proposes a novel adaptive cost strategy called the Gradient Confident Region (GradCoRe), which leverages the gradient's posterior uncertainty provided by the GP. At each SGD step, the GradCoRe strategy solves for the minimum number of measurement shots needed to ensure the gradient's uncertainty falls below a dynamically updated accuracy threshold. Numerical experiments show this combined SGD-GradCoRe method accelerates optimization, as measured by the number of shots on the quantum side, and outperforms existing state-of-the-art methods.

**Strengths:**

This paper's primary contribution is building on recent work by Nicoli et al. and Anders et al. to adapt their concepts to a more general framework. It takes the Confident Region (CoRe) idea, which was previously applied to 1D subspace methods, and adapts it for the D-dimensional gradients used in SGD, creating the Gradient Confident Region (GradCoRe).

The quality of this adaptation is well-supported on two fronts: theoretically, it provides a solid proof (Theorem 3.1) that its Bayesian PSR estimator is a valid generalization of the standard Parameter Shift Rule, and empirically, it demonstrates superior performance against strong, state-of-the-art baselines, including SubsCoRe from Anders et al that can be seen as a precursor model to the current one. This leads to a reduction of  the total number of required measurement shots by approximately 10-20% to reach the same energy levels in the benchmarks. The paper also maintains a high degree of clarity, using its experiments (Fig. 3) to effectively untangle the two sources of its performance gain. The results in Figure 3 indicate that while data-reuse alone (Bayes-SGD) provides no benefit over standard SGD, the entire performance lift is achieved by the adaptive measurement strategy (GradCoRe), which uses the GP's uncertainty to minimize shot cost.

Its significance comes from successfully broadening the utility of these physics-informed, adaptive-cost Bayesian methods beyond specialized algorithms, making them accessible to the widely-used SGD framework

**Weaknesses:**

The paper's novelty is somewhat limited, as its main contributions are largely an incremental adaptation of the VQE kernel  and Confident Region (CoRe) concept  developed by Nicoli et al. and Anders et al. While adapting this to the more general SGD framework is a valuable step, it relies heavily on these recently established foundations.

The paper's primary ablation study (Figure 3) reveals that Bayes-SGD, which incorporates the Bayesian PSR and data reuse, offers no clear performance benefit over standard SGD. This suggests that the paper's performance gain is derived exclusively from the GradCoRe adaptive shot-counting strategy. This undermines the presented value of the Bayesian PSR framework itself.

Additionally, a limitation in the empirical validation is the inconsistent use of the gamma kernel hyperparameter; Bayes-SGD use gamma=1 while all key baselines are set to gamma=3. The paper lacks an ablation study to quantify the performance sensitivity of all methods to this parameter, making it difficult to fully isolate the architectural gains of GradCoRe and Bayes-SGD from a potentially more favorable hyperparameter setting.

Lastly a very minor thing. I would argue that the paper overstates the novelty of Theorem 3.2, which proves the optimality of the π/2 shift. The proof primarily serves as a validation that the Bayesian framework aligns with existing knowledge, rather than being a new discovery.

**Questions:**

Figure 3  shows that Bayes-SGD (with data reuse) performs no better than standard SGD. This implies the performance gain of GradCoRe comes from its adaptive shot strategy, not from the Bayesian PSR or data reuse. Do the authors agree? If so, should the paper be re-framed to de-emphasize the Bayesian PSR and focus on GradCoRe as an adaptive shot-counting heuristic that can be applied to any PSR?

Please justify the use of a different kernel ($\gamma=1$) for GradCoRe versus the baselines ($\gamma=3$). This is a critical confounding variable that is not mentioned in the main text. Does GradCoRe still outperform SubsCoRe if both use $\gamma=3$? Or if both use $\gamma=1$?"

Since the practical contribution of this work is the non-insignificant reduction in number of shots, can you please expand a bit on the computational overhead (on the classical side) introduced by your model. Presumably this should scale as $\mathcal{O}(N^3)$ and you have chosen N=400 as mentioned in the appendix. The major drawback that VQE suffers from in terms of scalability is the amount of time required for training a model with a substantial number of parameters. A discussion on this computational tradeoff could really help your work.

---

> ### Author Response · Authors · 2025-11-21
> **Author response to Reviewer pQ8e**
>
> We thank the reviewer for their careful assessment of our manuscript and for the insightful questions raised.  Please find our answers below.  We also uploaded a revised manuscript, addressing reviewers’ comments (the revised/added sentences are highlighted in blue)
>
> >The paper's novelty is somewhat limited, as its main contributions are largely an incremental adaptation of the VQE kernel and Confident Region (CoRe) concept developed by Nicoli et al. and Anders et al. While adapting this to the more general SGD framework is a valuable step, it relies heavily on these recently established foundations.
>
> Although our method integrates elements from existing approaches, this should not be taken to undermine the novelty of our contribution. Theorems 3.1 and 3.2 introduce the theoretical basis required for Bayesian PSR, which represent a substantial novelty in itself.
>
> >The paper's primary ablation study (Figure 3) reveals that Bayes-SGD, which incorporates the Bayesian PSR and data reuse, offers no clear performance benefit over standard SGD. This suggests that the paper's performance gain is derived exclusively from the GradCoRe adaptive shot-counting strategy. This undermines the presented value of the Bayesian PSR framework itself.
>
> >Figure 3 shows that Bayes-SGD (with data reuse) performs no better than standard SGD. This implies the performance gain of GradCoRe comes from its adaptive shot strategy, not from the Bayesian PSR or data reuse. Do the authors agree? If so, should the paper be re-framed to de-emphasize the Bayesian PSR and focus on GradCoRe as an adaptive shot-counting heuristic that can be applied to any PSR?
>
> The advantage of Bayesian PSR is two folds: reuse of observations and uncertainty estimation. The data reuse certainly improves the GP prediction itself as seen in the improved gradient estimator in Fig. 7 in Appendix F.  Furthermore, data reuse seems to significantly improve the SMO-based method as seen in Fig.5 in Appendix B.  On the other hand, as the reviewer correctly pointed out, we do not observe performance gain by Baye-SGD with the data reuse in Fig. 3.  We will investigate why Bayes-SGD with a more accurate gradient estimator does not improve the optimization performance.
>
> Nevertheless, we still argue that Bayesian PSR is an indispensable component of the improved performance by GradCoRe, because GradCoRe relies on the uncertainty estimator provided by Bayesian PSR.  Therefore we respectfully disagree with the reviewer’s statement, and still believe the paper should discuss the Bayesian PSR.  We emphasized this point in Section 5.2 of the revised manuscript.
>
> Although the standard PSR does not provide uncertainty estimation, one may argue that the uncertainty could be estimated by error propagation.  However, data reuse by Bayesian PSR also contributes to the reduction of measurement shots by GradCoRe.  Namely, GradCoRe with more accurate gradient estimator by data reuse can reduce the number of shots much more than some shot adaptation method based on an uncertainty estimator for the standard PSR, because the uncertainty without data reuse should be larger than the uncertainty of Bayesian PSR.
>
> >Additionally, a limitation in the empirical validation is the inconsistent use of the gamma kernel hyperparameter; Bayes-SGD uses $\gamma = 1$ while all key baselines are set to $\gamma = 3$. The paper lacks an ablation study to quantify the performance sensitivity of all methods to this parameter, making it difficult to fully isolate the architectural gains of GradCoRe and Bayes-SGD from a potentially more favorable hyperparameter setting.
>
> >Please justify the use of a different kernel ($\gamma = 1$) for GradCoRe versus the baselines ($\gamma = 3$). This is a critical confounding variable that is not mentioned in the main text. Does GradCoRe still outperform SubsCoRe if both use $\gamma = 3$? Or if both use $\gamma = 1$ ?"
>
> The kernel width for GradCoRe is $\gamma = 3$ (see Table 2 in Appendix D.2 of the original and revised manuscripts), which is the same as for the other baselines, except the Bayes-SGD ($\gamma = 1$).  Therefore, the comparison with SubsCoRe for $\gamma = 3$ is already shown in Fig. 4.  Nevertheless, we agree that it is a confounding variable when GradCoRe and Bayes-SGD are compared.  We therefore replaced the results of Bayes-SGD in Fig.3 with $\gamma = 3$, and updated Table 2.  Namely, in the revision, all methods with GP are evaluated with $\gamma = 3$.

---

> ### Author Response · Authors · 2025-11-21
> **Author response 2 to Reviewer pQ8e**
>
> >Since the practical contribution of this work is the non-insignificant reduction in number of shots, can you please expand a bit on the computational overhead (on the classical side) introduced by your model. Presumably this should scale as O(N^3) and you have chosen N=400 as mentioned in the appendix. The major drawback that VQE suffers from in terms of scalability is the amount of time required for training a model with a substantial number of parameters. A discussion on this computational tradeoff could really help your work.
>
> In the revision, we discuss the classical computation cost in Appendix H.  As the reviewer correctly mentioned, we need to perform GP predictions, with at most $N=400$ training points, $D$ times in each SGD iteration.  As shown in Table 4, the wall-clock time per optimization step is $\sim 1 \cdot 10^{-5}$ sec for SGD (with the standard PSR), and is $\sim 4 \cdot 10^{-3}$ sec for Bayes-SGD and GradCoRe, i.e., the classical computation time of GradCoRe is 400 times larger than the SGD.  However, the classical computation time is still small and reasonable to be ignored in the context of VQE optimization.
>
> >Lastly a very minor thing. I would argue that the paper overstates the novelty of Theorem 3.2, which proves the optimality of the π/2 shift. The proof primarily serves as a validation that the Bayesian framework aligns with existing knowledge, rather than being a new discovery.
>
> Although $\pi/2$ shift for PSR is commonly used, to the best of our knowledge there are no works which discuss its optimality. Therefore, our Theorem 3.2 is the first theoretical result that proves such optimality.  Actually, $\pi/2$ shift is also widely used for SMO (or NFT), which was recently proven suboptimal in ["Optimal parameter configurations for sequential optimization of the variational quantum eigensolver"](https://link.aps.org/doi/10.1103/PhysRevResearch.5.043136). Together with Theorem 3.1, Theorem 3.2 proves the optimality for the original PSR, which is equivalent to Bayesian PSR when $ \sigma^2 \ll \sigma_0$.

---

> > ### Comment · Reviewer_pQ8e · 2025-11-27
> > **Response to authors**
> >
> > Dear authors,
> >
> > thank you for taking the time to give such a detailed response to my questions/concerns. I largely agree with the points you just raised. In light of these clarifications and the revisions made to the manuscript, I believe the paper now meets the acceptance bar. I have raised my rating to 8.
> > Best,

---

### Author Response · Authors · 2025-11-21
**General response**

We thank the reviewers for their encouraging evaluations and constructive suggestions.  We uploaded a revised version of our manuscript, reflecting the reviewers’ comments.  The revised/added sentences, highlighted in blue, consist of

- A derivation of the derivative kernels for GP in Section 2 (suggested by Reviewer 9nAn).

- Explanation for the reason for the “U” shape curve observed in Fig. 2 right (suggested by Reviewer TZCS).


- Fig. 3 is replaced (as suggested by pQ8e).  Now all methods use the same kernel parameter $\gamma = 3$ .

- Emphasizing the importance of Bayesian PSR for GradCoRe (addressing the comments by Reviewer pQ8e).

- Classical computation cost discussion in Appendix H (suggested by Reviewers pQ8e and 9nAn) .

- Gaussianity of the observation noise is validated in Appendix I (addressing the comments by Reviewers TZCS and  9nAn).

Our response to each review is given as an official comment to the corresponding reviewer  below.

---

### Meta-Review · Area_Chair_78vk · 2026-01-06

**Summary:**

The paper proposes a Bayesian variant of the Parameter Shift Rule (PSR) using Gaussian Processes (GPs) with a physics-informed kernel. The primary contributions include a theoretical framework for data reuse in gradient estimation and the Gradient Confident Region (GradCoRe) strategy, which adaptively allocates measurement shots based on posterior uncertainty.

**Reviewer Concerns:**

Reviewers generally appreciated the theoretical grounding (Theorems 3.1 and 3.2) and the empirical performance gains in reducing quantum measurement costs. While initial concerns were raised regarding novelty and hyperparameter consistency, the authors'rebuttal convinced the most active reviewer to significantly raise their score.

Specifically, Reviewers fLQm and pQ8e initially felt the work was an incremental step over prior "CoRe" methods. The authors addressed this by highlighting that Theorem 3.2 provides the first formal proof of the optimality of the $\pi/2$ shift.

Reviewer pQ8e noted inconsistent $\gamma$ settings. The authors provided a revised Fig. 3 using uniform hyperparameters, which satisfied the reviewer.

Reviewer 9nAn and pQ8e questioned the classical computational cost of GP regression. The authors added Appendix H, demonstrating that while the cost is higher than standard SGD, it remains negligible (${\sim}4 \times 10^{-3}$ sec) compared to quantum evaluation time.

**Reviewer Scores:**

Reviewer pQ8e: Changed from 6 to 8 (confirmed in discussion).

Reviewer TZCS: 6 (likely maintained; concerns on intuition/GP justification were addressed).

Reviewer 9nAn: 8 (highly positive initially; concerns on overhead and derivation were addressed).

Reviewer fLQm: 4 (no response to rebuttal, but concerns regarding motivation and novelty were addressed by the authors).

---

### Decision · Program_Chairs · 2026-01-26

Accept (Poster)